# Reduced Atlantic reef growth past 2 °C warming amplifies sea-level impacts

Chris T. Perry[1 ✉], Didier M. de Bakker[1], Alice E. Webb[1], Steeve Comeau[2], Ben P. Harvey[3], Christopher E. Cornwall[4], Lorenzo Alvarez-Filip[5,6], Esmeralda Pérez-Cervantes[5], John Morris[7], Ian C. Enochs[8], Lauren T. Toth[9], Aaron O'Dea[10,11], Erin M. Dillon[10], Erik H. Meesters[12] & William F. Precht[13]

Coral reefs form complex physical structures that can help to mitigate coastal flooding risk[1,2]. This function will be reduced by sea-level rise (SLR) and impaired reef growth caused by climate change and local anthropogenic stressors[3]. Water depths above reef surfaces are projected to increase as a result, but the magnitudes and timescales of this increase are poorly constrained, which limits modelling of coastal vulnerability[4,5]. Here we analyse fossil reef deposits to constrain links between reef ecology and growth potential across more than 400 tropical western Atlantic sites, and assess the magnitudes of resultant above-reef increases in water depth through to 2100 under various shared socioeconomic pathway (SSP) emission scenarios. Our analysis predicts that more than 70% of tropical western Atlantic reefs will transition into net erosional states by 2040, but that if warming exceeds 2 °C (SSP2–4.5 and higher), nearly all reefs (at least 99%) will be eroding by 2100. The divergent trajectories of reef growth and SLR will thus magnify the effects of SLR; increases in water depth of around 0.3–0.5 m above the present are projected under all warming scenarios by 2060, but depth increases of 0.7–1.2 m are predicted by 2100 under scenarios in which warming surpasses 2 °C. This would increase the risk of flooding along vulnerable reef-fronted coasts and modify nearshore hydrodynamics and ecosystems. Reef restoration offers one pathway back to higher reef growth[6,7], but would dampen the effects of SLR in 2100 only by around 0.3–0.4 m, and only when combined with aggressive climate mitigation.

Sea-level rise (SLR) is projected to increase the frequency of coastal inundation globally, threatening coastal habitats, human coastal communities and infrastructure[8,9]. In addition, SLR will increase coastal wave exposure in locations where ecological degradation compromises the wave-attenuating capabilities of nearshore habitats, such as seagrass beds, salt marshes and mangroves[10–12]. Nearshore coral reefs can also fulfil this wave-protecting role by dissipating wave energy across shallow reef zones[13]. Globally, reef-derived coastal protection functions benefit an estimated 5.3 million people and protect coastal assets valued at around US$109 billion per decade[1]. However, persistence of this key functional role requires that reef growth (hereafter, maximum reef accretion potential, or RAP$_{max}$)[3,14] across the shallower regions of reefs closely tracks SLR[15]. Where RAP$_{max}$ rates lag behind SLR, water depths increase, changing across-reef wave heights and wave energy transfer[16,17]. Sustained reef accretion that could balance the effects of SLR can occur only if reef calcifying taxa generate more

skeletal carbonate than is lost to the processes of physical, chemical and biological erosion. The balance between these processes is described as a reef's carbonate budget ($G$, where $G$ = kg CaCO$_3$ m$^{-2}$ yr$^{-1}$). Anthropogenically induced changes to reef ecosystems have already caused widespread declines in reef carbonate budgets[14,18–20] and thus in RAP$_{max}$. The low $G$ values reported on reefs across the tropical western Atlantic (TWA) ($2.55 \pm 3.83$ (mean ± s.d.)) and Indian Ocean ($1.41 \pm 3.02$) regions have, for example, been estimated to equate to RAP$_{max}$ rates of only $1.87 \pm 2.16$ mm yr$^{-1}$ and $2.01 \pm 2.33$ mm yr$^{-1}$, respectively. These rates are well below average recent SLR rates[21] (3.6 mm yr$^{-1}$), suggesting that reef accretion is already lagging behind SLR on many reefs[3].

Climate change represents a severe and magnifying threat to reef carbonate budgets[22] and to reef accretion which will worsen climate-driven SLR effects. Coral bleaching caused by thermal stress has already affected coral cover and diversity in many locations, often exacerbated by disease outbreaks[23], resource over-extraction and poor

[1]Geography, Faculty of Environment, Science and Economy, University of Exeter, Exeter, UK. [2]Sorbonne Université, CNRS-INSU, Laboratoire d'Océanographie de Villefranche, Villefranche-sur-Mer, France. [3]Shimoda Marine Research Center, University of Tsukuba, Shimoda, Japan. [4]School of Biological Sciences, Victoria University of Wellington, Wellington, New Zealand. [5]Biodiversity and Reef Conservation Laboratory, Unidad Académica de Sistemas Arrecifales, Instituto de Ciencias del Mar y Limnología, Universidad Nacional Autonoma de México, Puerto Morelos, Mexico. [6]Laboratorio Nacional de Biología del Cambio Climático, SECIHTI, Ciudad de México, Mexico. [7]Cooperative Institute for Marine and Atmospheric Studies, University of Miami, Miami, FL, USA. [8]Atlantic Oceanographic and Meteorological Laboratory, Ocean Chemistry and Ecosystem Division, NOAA, Miami, FL, USA. [9]U.S. Geological Survey, St Petersburg Coastal and Marine Science Center, St Petersburg, FL, USA. [10]Smithsonian Tropical Research Institute, Balboa, Republic of Panama. [11]Sistema Nacional de Investigación, SENACYT, Panama, Republic of Panama. [12]Wageningen Marine Research (WMR), Wageningen University & Research, Den Helder, The Netherlands. [13]Bio-Tech Consulting, Coastal and Marine Sciences, Miami Lakes, FL, USA. ✉e-mail: c.perry@exeter.ac.uk

water quality[24]. Future thermal stresses are predicted to accelerate coral loss, and ocean acidification will progressively reduce coral and crustose coralline algal (CCA) calcification and increase substrate erosion[25]. Accurately predicting emerging risks to tropical shorelines and nearshore ecosystems from SLR will require a better understanding of how $RAP_{max}$ rates will respond to (i) future environmentally driven changes in coral cover (a key determinant of reef carbonate production rates) and (ii) interactive effects of sea-surface temperature (SST) and ocean acidification on calcification and bioerosion rates.

Here we tackle these challenges. First, we use an analysis of palaeo-reef deposits (Extended Data Fig. 1) to address a major area of uncertainty[20] in previous $RAP_{max}$ projections, which was caused by poor constraints on how reef framework stacking porosity varies with coral assemblage (see Methods). Stacking porosity is the vertical space added for a given volume of carbonate produced by an assemblage of corals, and is a key metric used in converting reef carbonate production data to estimates of reef accretion (see Methods). We then quantify contemporary $RAP_{max}$ rates in three geographically distinct locations across the TWA using the new assemblage-specific conversion factors. We focus on back-reef to shallow fore-reef habitats because in the TWA, where reef flat habitat is rare, these zones are the most important for wave energy dissipation[6,26,27]. We then examine how $RAP_{max}$ will change through to 2100 under various SSP emission scenarios (Supplementary Table 1) by factoring for effects on coral cover, carbonate calcification and substrate bioerosion. Finally, we quantify $RAP_{max}$ in relation to projected local SLR rates (mm yr$^{-1}$) for each SSP scenario[28]. We use this analysis to investigate variations in above-reef water depth increases by 2040, 2060 and 2100 under each SSP. This analysis enabled us to identify whether and when the shallowest areas of reefs in three TWA subregions would experience water-level increases exceeding 0.5 m. We take 0.5 m as an indicative threshold beyond which increased coastal wave exposure and flooding risk commonly occurs[16,17,27].

## Revising estimates of reef accretion rate

Our calculation of reef framework porosities is based on image analysis of fossil coral assemblages (Extended Data Fig. 2) and shows a mean (±s.d.) framework porosity across coral assemblages of 38.3 ± 10.1%. This is substantially lower than the 50% mean proposed in previous studies[29,30]. Notably, porosity values diverged markedly for specific coral assemblages (Fig. 1a and Supplementary Table 2). For assemblages dominated by branching-morphology corals, including *Acropora palmata* (36.9 ± 3.4%), *Acropora cervicornis* (55.4 ± 3.9%) and *Porites* spp. (47.3 ± 8.7%), porosities were much lower than the 70–80% values originally suggested for Pacific branching-coral communities[30] (Fig. 1a). Calculated porosity values (25–35%) for communities composed of massive-morphology corals were similar to those originally recommended[30]. Collectively, the use of these revised lower porosity values reduces accretion estimates, highlighting the need to reassess contemporary reef growth estimates to more reliably predict the effects of current and future SLR.

Using this revised understanding of assemblage-specific porosity values, we performed a comparative analysis against previous Caribbean $RAP_{max}$ data[3]. The net effect across the regional dataset was a modest but significant ($P < 0.001$) decline: 12.4% on average across all sites (from 1.87 ± 2.16 to 1.64 ± 1.76 mm yr$^{-1}$ (mean ± s.d.); Fig. 1b and Supplementary Table 3). There were significant declines at the subregion level in the Leeward Antilles (−27.4%; 4.87 ± 2.71 to 3.54 ± 1.96 mm yr$^{-1}$; $P < 0.001$) and the Mesoamerican reef (−19.4%; 0.63 ± 1.39 to 0.51 ± 1.31 mm yr$^{-1}$; $P < 0.05$) (Fig. 1b). The most significant reductions in $RAP_{max}$ occurred in sites with the highest coral cover (Fig. 1c) and carbonate budgets (Fig. 1d). The divergent effect of using these revised porosity values is especially evident when comparing sites with different coral assemblages. $RAP_{max}$ rates are more than halved ($P < 0.01$) where sizeable communities of branching *Acropora* spp. corals persist (Fig. 1e).

Conversely, the effect was limited at sites with low cover of mainly encrusting and massive-morphology corals and with marginal or negative carbonate budgets (Fig. 1e).

## Contemporary estimates of reef accretion rate

Given the sensitivity of $RAP_{max}$ rates to assemblage-specific porosity factors, we analysed the three largest existing datasets on TWA carbonate budgets to investigate how contemporary $RAP_{max}$ rates vary across and within TWA subregions. This analysis addresses a key limitation of previous studies[3], which had low in-country site replication. We analysed datasets collected between 2016 and 2022, encompassing sites along the Florida Keys ($n = 113$), the Mexican Mesoamerican reef and in the Mexican sector of the Gulf of Mexico ($n = 88$), and around Bonaire ($n = 228$). Our data show that mean (±s.d.) $RAP_{max}$ rates are close to net neutral or are slightly net negative—that is, in net erosional states on average—across the Florida Keys (−0.06 ± 0.40 mm yr$^{-1}$; 61.9% of sites net eroding) and Mexican sites (0.28 ± 1.22 mm yr$^{-1}$; 38.4% of sites net eroding), and only slightly higher on average in Bonaire (0.91 ± 1.43 mm yr$^{-1}$; 29.8% of sites net eroding; Fig. 2). Such rates are substantially below average long-term (Holocene) western Atlantic reef accretion rates[31] (4.8 mm yr$^{-1}$). We also note clear differences in the range of $RAP_{max}$ rates calculated within each subregion (Extended Data Fig. 3). All Florida Keys sectors are defined by uniformly very low net positive or net negative $RAP_{max}$ values (range: 1.40 to −1.11 mm yr$^{-1}$; Fig. 2). This is indicative of the general poor ecological health of the entire Florida Keys[20], as demonstrated by the absence of branched *Acropora* colonies at all survey locations. By contrast, outlier sites of higher coral cover and/or with communities of high-rate carbonate-producing *Acropora* taxa persist in both of the other subregions and are associated with higher $RAP_{max}$ rates (Mexican sites up to 8.01 mm yr$^{-1}$; Bonaire up to 6.89 mm yr$^{-1}$; Fig. 2 and Extended Data Fig. 3).

Our revised $RAP_{max}$ data indicate that no contemporary reef sites in Florida and only a few in Mexico (5.5%) and Bonaire (5.7%) persist with $RAP_{max}$ rates that exceed rates of SLR from 1993 to 2010 (Fig. 2). Comparisons to near-future (2040) SLR rates under even the lowest $CO_2$ emission scenario (SSP1–1.9) are even more concerning, with only one site in Mexico (a site which retains high *A. palmata* cover) and five sites in Bonaire having $RAP_{max}$ rates that exceed SSP1–1.9 SLR projections (Fig. 2). The current ecological state of TWA reefs thus suggests limited capacity to track even modest SLR. This situation is more severe than suggested by previous estimates, in which around 45% of reefs had $RAP_{max}$ rates close to (within ±1 mm yr$^{-1}$) or higher than recent local (altimetry-derived) SLR rates[3]. Above-reef water depth levels are therefore likely to increase higher and faster than previously anticipated.

## Climate change and future reef accretion

Ongoing background climate warming, more frequent and severe ocean thermal stress events (marine heatwaves) and ocean acidification effects will further impair the capacity of reefs to track future SLR by reducing $RAP_{max}$. These effects will arise from (i) increased rates of coral mortality, leading to projected coral cover levels close to zero under all SSP scenarios by around 2080 (Extended Data Fig. 4) and (ii) suppressed coral and coralline algal calcification and increased substrate bioerosion resulting from both warming and ocean acidification (Supplementary Table 4). TWA reef carbonate budget states and resulting $RAP_{max}$ estimates towards the later parts of this century will thus be progressively driven by bioerosion as opposed to coral and CCA calcification. Our analysis of future changes to $RAP_{max}$ under these climate stressors shows a progressive decline in $RAP_{max}$ through to 2100 across all SSP scenarios and in each subregion (Fig. 3a and Extended Data Fig. 5). Even under the optimistic SSP1–2.6 emission scenario (with warming staying below 2 °C), a high percentage of reef sites will be net eroding by 2040 (Florida 78.7%, Mexico 87.5% and Bonaire 68.9%).

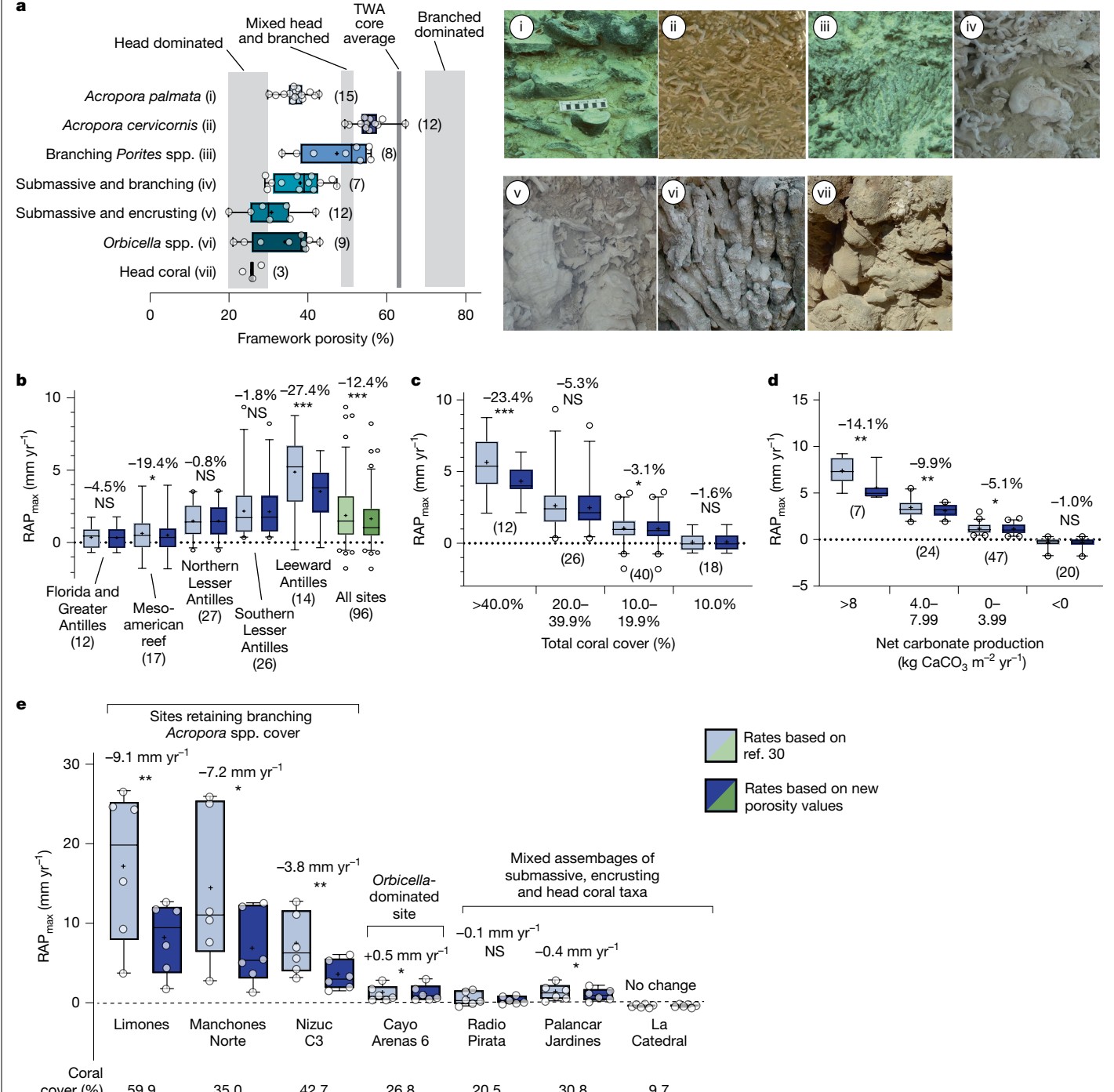

**Fig. 1 | Calculated framework porosities for western Atlantic coral assemblages and impacts on reef accretion potential. a,** Framework porosity (%) for the seven types of coral assemblage examined (example images i–vii), as delineated by the dominant species or morphologies present. Vertical light-grey bands show proposed stacking porosity values for head-dominated, mixed head-and-branching-dominated and branching-dominated coral reef sequences proposed in previous studies[30]. The dark-grey vertical line is the mean porosity value determined from TWA reef cores[20]. **b–e,** Comparisons of calculated RAP_max rates using original framework porosity values[30] with those calculated here and applied to existing TWA reef budget data[3]. These consider differences across all TWA sites (**b**); differences between sites grouped into coral cover classes (**c**); differences between sites grouped into total net carbonate production classes (**d**); and differences across a range of sites ($n = 6$ per site) in Mexico that differ in their coral cover and coral assemblage type (**e**). Box plots depict median (horizontal line), mean (black cross) and first and third quartiles (box limits). Whiskers represent the 95th percentile. Sites outside the 95th percentile are shown as circles. Numbers above the plots indicate the change from previous estimates. Significance levels are from paired $t$-tests: NS, not significant ($P > 0.05$); *$P < 0.05$, **$P < 0.01$, ***$P < 0.001$. Numbers in parentheses are replicates per dataset.

Under higher-emission scenarios (SSP3–7.0 and SSP5–8.5), and by 2060, the percentage of reef sites in net erosional states further increases (Florida 92.0% and 98.0% respectively; Mexico 100.0% for both scenarios; Bonaire: 70.6% and 79.4%; Fig. 3a), and by 2100, mean RAP_max rates for each subregion and under each emission scenario are projected to be net negative. Projected mean (±s.d.) RAP_max rates (mm yr⁻¹) at 2100 range from −0.27 ± 0.20 (SSP1–2.6) to −0.35 ± 0.30 (SSP5–8.5) in Florida; from −0.69 ± 0.65 (SSP1–2.6) to −0.91 ± 0.65 (SSP5–8.5) in

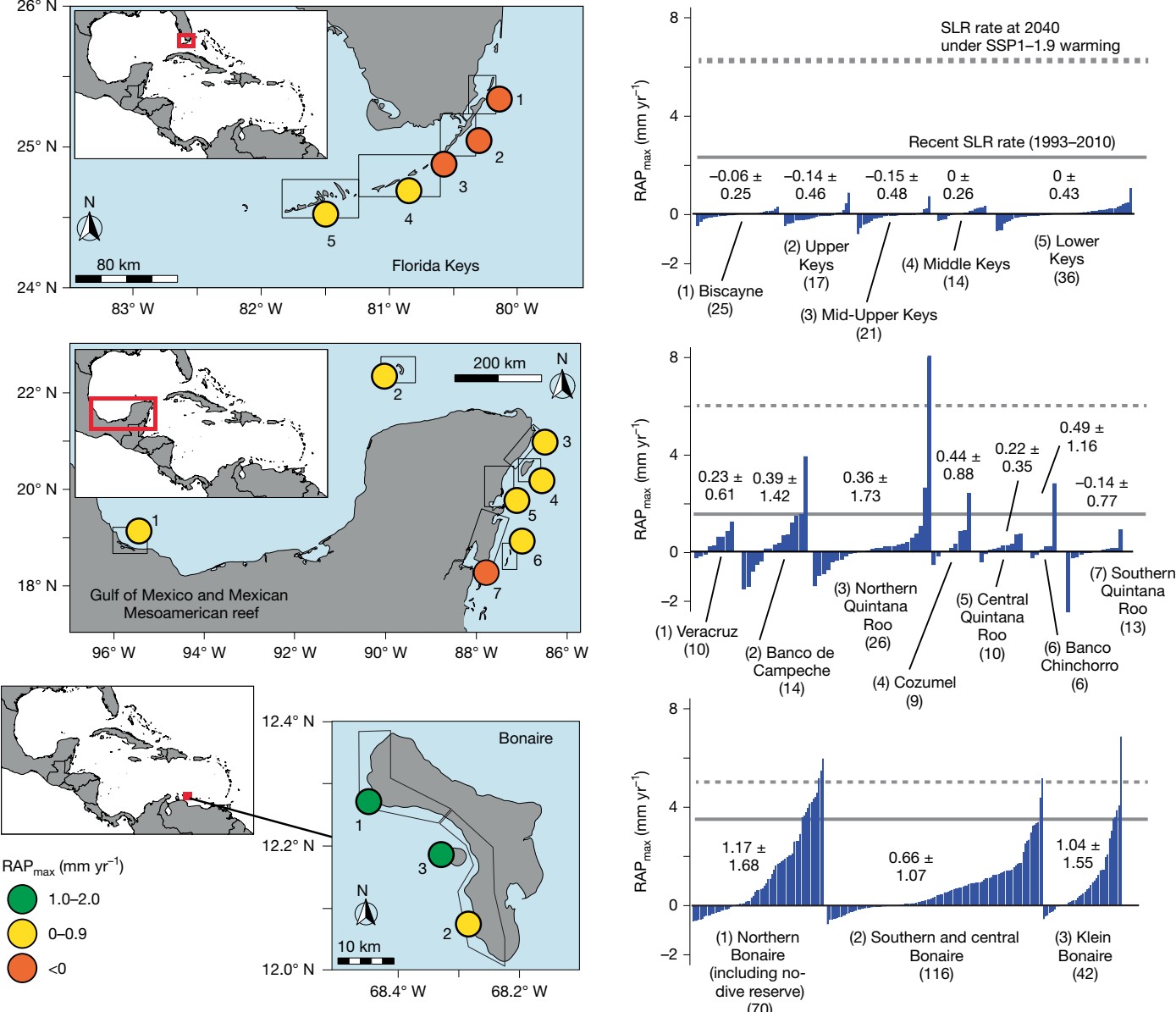

**Fig. 2 | Reef accretion potential across western Atlantic reefs.** Left, maps showing the location of reef sectors in each study subregion. Coloured circles denote the mean $RAP_{max}$ rate (mm yr$^{-1}$) for each sector, and the numbers refer to the subsectors listed in the plots on the right. Right, contemporary mean (±s.d.) $RAP_{max}$ rates (mm yr$^{-1}$) for each study subregion, at site level and grouped by sector. Numbers in parentheses are replicates per dataset. Data are compared with SLR rates for the period 1993–2010 for each subregion (solid grey line) and SLR rates projected at 2040 under SSP1–1.9 (dashed grey line), under which the global warming range at 2100 is projected to be 1.0–1.8 °C.

Mexico; and from −0.41 ± 0.32 (SSP1–2.6) to −0.63 ± 0.34 (SSP5–8.5) in Bonaire (Fig. 3a). Overall magnitudes of decline are slightly greater by 2100 in Mexico and Bonaire, because in our projections we necessarily assume continued higher rates of biological erosion in these locations compared with Florida, where rates are relatively low[32]. Of the 429 sites that we consider, only two sites in Bonaire and one in Florida persist with low net positive $RAP_{max}$ rates at 2100 under SSP2–4.5, and none do so in any location under the higher-emission scenarios (Fig. 3a).

A major consequence of these transitions into net low and negative (that is, net erosional) $RAP_{max}$ states will be that the capacity of many TWA reefs to sustain coastal protective functions under SLR is progressively compromised. SLR rates are projected to increase rapidly[28] through to 2100 (Supplementary Table 7) and thus the two main controls on above-reef water depths—the rate of reef accretion and the rate of SLR—will increasingly operate in divergent directions. This will magnify SLR effects. Our projected $RAP_{max}$ rates (which are

conservatively high; see Methods), when compared against future SLR projections (Supplementary Table 7), indicate that water depths above both shallower (0–6 m) and deeper (6–12 m) habitats of TWA reefs will increase by an average of 0.3–0.5 m (depending on SSP scenario) by 2060 (Fig. 3b and Extended Data Fig. 6), and will accelerate rapidly between 2060 and 2100, reflecting increasing SLR rates (Supplementary Table 7). Mean (95% confidence interval (CI)) projected water depth increases at 2100 under SSP2–4.5 and SSP5–8.5 are 0.71 m (0.33–1.33) and 1.09 m (0.57–2.05) in Florida, 0.76 m (0.30–1.43) and 1.20 m (0.61–2.14) in Mexico and 0.65 m (0.19–1.33) and 1.04 m (0.50–1.95) in Bonaire. Differences between subregions mostly reflect differences in rates of SLR (Supplementary Table 7). The result will be that most reefs will progressively experience more than 0.5 m of water depth increase relative to reef topography. This would approximately double mean water levels above the shallowest areas of many TWA reefs, leading to a decrease in depth-limited wave breaking and bottom friction[2,16,17].

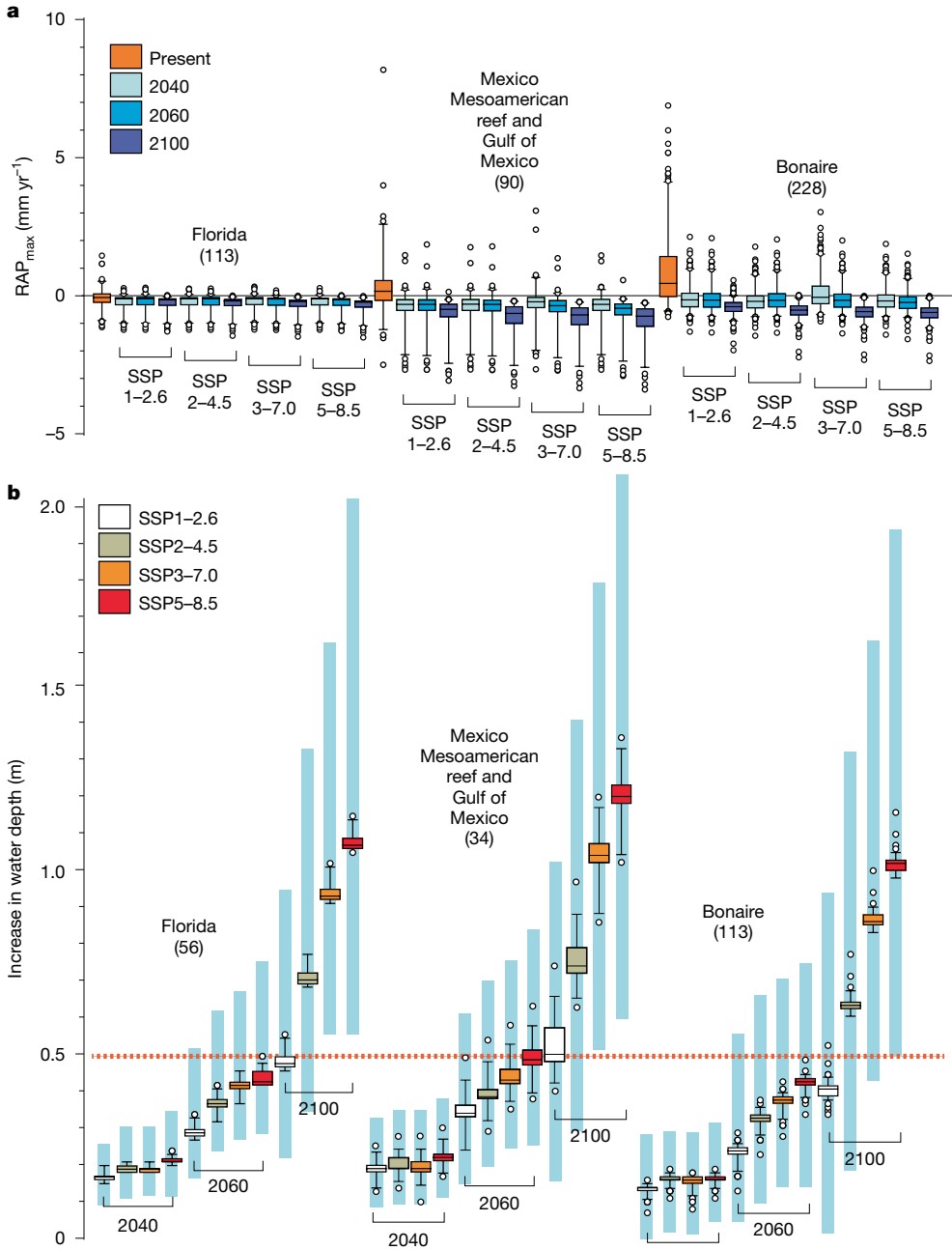

**Fig. 3 | Projected changes in reef accretion potential and resulting increases in water depth at 2040, 2060 and 2100 under various SSP emission scenarios. a**, $RAP_{max}$ rates (mm yr$^{-1}$) at present, and at 2040, 2060 and 2100 under various SSP scenarios (SSP1–2.6, SSP2–4.5, SSP3–7.0 and SSP5–8.5) in each subregion. The data from Mexico integrate sites that span two ecoregions: the western Caribbean and the southern Gulf of Mexico. **b**, Projected increases in total water depth (m) above back-reef to shallow reef front (0–6 m depth) zones of reefs in each subregion under SSP scenarios at 2040, 2060 and 2100. Box plots depict median (horizontal line) and first and third quartiles (box limits). Whiskers represent the 95th percentile. Sites outside the 95th percentile are shown as circles. In **b**, the pale-blue vertical bars behind each box show the total range of mean projections of water depth increase based on lower (5%) and upper (95%) CI projections for the interactive effects of coral cover change and rate of SLR under each SSP scenario. Numbers in parentheses are replicates per scenario and time point. The dashed red horizontal line in **b** represents 0.5 m of additional water depth increase, above which substantially increased coastal wave energy exposure and flooding risks are likely[16,17].

## Implications and mitigation options

Our findings project a bleak future for the capacity of TWA coral reefs to limit SLR effects under future climate scenarios. Even under the optimistic SSP1–2.6 scenario, which limits global warming to below 2 °C by 2100, mean water depth increases above reefs are projected to be close to 0.5 m (Florida 0.48 m, Mexico 0.52 m and Bonaire 0.41 m; Fig. 3b), raising concerns about the future effectiveness of TWA reefs in mitigating coastal wave energy exposure and limiting flooding risks in vulnerable locations[17,33]. More-extreme—yet plausible—warming scenarios (for example, SSP3–7.0, under which mean water depths would increase by around 0.95 m by 2100), would magnify these negative outcomes. In addition, unforeseen ecological consequences might arise in which increased wave-overtopping enhances the exchange of water and sediment between lagoons and the open ocean[34]. Most concerning is that much of this projected water depth increase arises from the magnitude

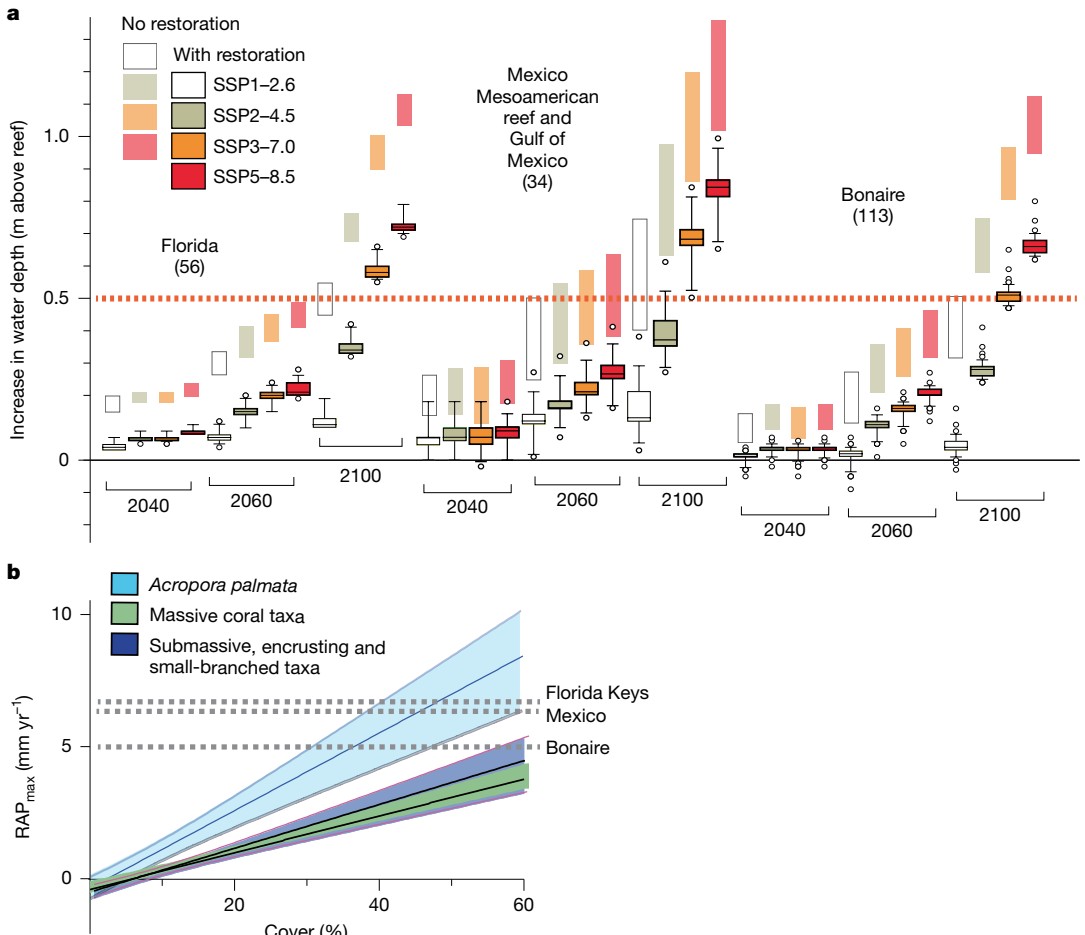

**Fig. 4 | Projected rates of reef accretion potential for restored coral communities compared with rates based on contemporary projections, and relationships between accretion rate and coral cover. a**, Box plots showing projected increases in total water depth (m) above back-reef to shallow reef front (0–6 m depth) zones of reefs in each subregion under SSP scenarios at 2040, 2060 and 2100, recalculated on the basis of a highly successful and long-lasting restoration effort providing an additional 4.8 mm yr⁻¹ of reef accretion (see text for discussion). Numbers in parentheses are replicates per scenario and time point. Box plots depict median (horizontal line) and first and third quartiles (box limits). Whiskers represent the 95th percentile. Sites outside the 95th percentile are shown as circles. Dashed horizontal line represents 0.5 m of additional water depth increase. Pale-coloured boxes above box plots show, for comparison, the range (maximum–minimum value) from mean projections of water depth increase under each SSP scenario from Fig. 3b (without restoration). **b**, Relationships between cover (%) and calculated $RAP_{max}$ (mm yr⁻¹) for sites with *A. palmata* ($n = 27$) and for sites dominated by massive coral taxa ($n = 158$) or by submassive, encrusting and small-branched coral taxa ($n = 120$). Solid lines represent the fitted linear regression; shaded areas show the 95% CI of the mean regression line; *A. palmata*: $y = 0.1457x − 0.3347$ ($r^2 = 0.7515$); massive taxa: $y = 0.06977x − 0.4042$ ($r^2 = 0.5416$); submassive, encrusting and small-branched taxa: $y = 0.07871x − 0.5278$ ($r^2 = 0.3404$). Dashed lines show subregion-level SLR projections at 2040 under SSP1–1.9 (as in **a**).

of SLR itself. Many TWA reefs are now eroding (Supplementary Table 8) or accreting so slowly that they seem to have a limited capacity to respond positively to SLR through accelerated rates of carbonate production. This limited response capacity was further impaired by the widespread losses of remaining reef-building *Acropora* species during the 2023–2024 bleaching event[35]. What this means is that the long-term linkages and interplay that have occurred between reef accretion and SLR, and which have influenced TWA reef elevation with respect to sea level over millennia[36], will be severed. Whether these projections will hold or follow similar patterns in the Indo-Pacific remains unclear. Many reefs in that region have expansive reef flats, reflecting the region's different sea-level history, and SLR might open space for renewed vertical accretion[37,38]. This potential will clearly depend on the effects of future thermal stresses and other disturbance events on coral populations.

Reef restoration is one strategy for transitioning reefs back into more positive budget states, and thus for addressing the issue of reef accretion–SLR rate divergence[6,39]. Approaches based on coral outplanting are commonly challenged, however, not only by local environmental pressures and climate-related stressors such as coral mass bleaching,

but also by the simple fact of the spatial scales involved (TWA reefs cover an area of around 26,000 km²). Successful approaches to coral restoration at such scales do not yet exist[40], although recent studies have shown that, at local scales and with intensive and sustained effort (personnel and financial), restoration can deliver rapid gains to carbonate budgets and reef accretion[7,41]. Given this, it is informative to conceptualize what the most successful theoretical restoration outcomes might deliver on TWA reefs. For the reasons given above, such outplanting would be most effective at keeping pace with SLR if it is focused on the upper reef crest or back-reef zones of TWA reefs, because SLR will not open major areas of new colonizable habitat elsewhere.

One approach is to consider a scenario in which successful restoration yields net long-term accretion rates that are consistent with those that defined the region's reefs during the Holocene[31] (that is, around 4.8 mm yr⁻¹). If we assume that this is feasible, a simple application of these enhanced accretion rates across our datasets (which provide a broad regional assessment of TWA reef ecological performance) and projected through to 2100 shows that many more reefs, and for longer, would experience water depth increases of less than 0.5 m (Fig. 4a).

Indeed, not only would attaining such an accretion rate provide something of a climate-change buffer (only under SSP3–7.0 and SSP5–8.5 would most reefs experience more than 0.5 m of water depth increase), but some sites in Bonaire would also experience net shallowing under a warming scenario of less than 2 °C. A crucial question, however, is what minimum levels of coral cover would be needed to achieve such accretion rates. On the basis of a regression analysis of our calculated $RAP_{max}$ data against percentage cover levels for various common shallow-water TWA coral species and assemblages, sustained cover of at least 35–40% *A. palmata*, or 60–70% for communities dominated by massive, or by submassive, encrusting and small-branched taxa would be required to generate $RAP_{max}$ rates close to 5 mm yr$^{-1}$ (Fig. 4b). These cover levels seem extremely ambitious given the current state of restoration successes, and because, first, they exceed the cover reported on most reefs studied here before recent coral losses[24], and second, the highest-accreting communities (*Acropora* dominated) are also the most susceptible to environmental disturbances. Concerningly, even higher levels of cover would be needed for reefs to track SLR rates projected under all emission scenarios beyond 2040.

This is a simplified hypothetical example for illustrative purposes, and more nuanced modelling could be performed as empirical data improve to discern how different combinations of coral taxa under different planting densities and acclimation and survival outcomes[6,39] might modify $RAP_{max}$ rates. However, this example shows that if successful and high-density outplanting-based restoration could occur at scale, it could be possible to mitigate the worst effects of SLR[6]. At the very least, this might limit transitions towards more negative $RAP_{max}$ rates, buying time for coral acclimation or blue-economy transitions. Given the current state of many TWA reefs, any actions that generate additional framework building are helpful, not least because reef-derived coastal protection benefits are also supported by enhanced physical structural complexity, which can help to dissipate wave energy[2] and reduce extreme wave run-up[42]. Hybrid restoration combining coral outplanting on artificial structures could be an alternative, if costly, option[43]. Equally, natural or facilitated coral acclimation and adaptation that promotes coral thermal tolerance[44,45] could support pathways back to higher coral cover, carbonate budgets and reef accretion rates. Even with this occurring, the scale of restoration needed to restore TWA reefs is daunting (the Mesoamerican reef alone is around 1,000 km long), and its outcomes are uncertain, owing to high background mortality and low recruitment of outplanted corals[40,46]. Actions to tackle known local drivers of reef ecological decline such as water quality and overfishing[47,48] might help to partially address this. However, given an existing commitment to SLR caused by ocean thermal expansion and land ice melting, this study suggests that actions to keep warming below 2 °C are crucial to limiting SLR impacts along the region's reef-fronted coastlines.

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

## Methods

### Quantifying reef accretion potential

We calculated coral reef maximum vertical accretion rate potential (mm yr$^{-1}$) (hereafter RAP$_{max}$)[3,14] using a long-standing methodology first proposed by Smith & Kinsey[29] and then refined by Kinsey & Hopley[30]. This is based on the conversion of in-field measures of reef carbonate production ($G$, in kg CaCO$_3$ m$^{-2}$ yr$^{-1}$), using a mineral density of 2.9 g cm$^{-3}$ and by integrating a stacking porosity factor for the associated coral assemblage[30]. This now widely used method provides a conservatively high estimate of reef accretion because derived rates are necessarily, owing to a lack of empirical data, calculated without the inclusion of physical or chemical solution losses. Omission of the former is arguably less important at present on TWA reefs because of the paucity of more physically vulnerable coral taxa (branching corals)[49]. A fundamental aspect of this conversion approach is the use of a stacking porosity factor (the space added vertically for a given volume of carbonate produced by an assemblage of corals post-mortem and after any biological erosion or physical denudation). In relation to this, some coral morphotypes can be preserved more or less in growth position (massive growth forms), but others, such as branching corals, will typically be broken down by physical processes and accumulate as piles of fragmented rubble cemented together by coralline algae and secondary precipitation of CaCO$_3$. The nature of skeletal framework accumulation and its stacking porosity is thus influenced strongly by the types of coral (and their growth morphologies) present in a community or deposited in a site.

Early studies proposed a range of porosity values derived from Indo-Pacific reef-core data for application to assemblages dominated by branching, mixed or massive coral. Here, Kinsey & Hopley stated that "Higher porosities up to 80% may be associated with branching coral assemblages […] however, head corals produce denser framework [and] porosities can therefore vary between 20 and 80%…"[30]. Derived reef accretion rates are, however, highly sensitive to the use of slightly higher or lower porosity values. This implies that substantial improvements to resultant accretion rate estimates could be made if the stacking porosity values associated with specific coral assemblages could be constrained. For example, in the TWA it is reasonable to hypothesize that the stacking porosities of different types of branching-coral-dominated assemblages (for example, *A. palmata* versus *A. cervicornis* versus branching *Porites* species), each of which have very different branch thicknesses and geometries, and break down post-mortem in different ways, will vary sufficiently to influence resultant accretion estimates. Indeed, the potential for such differences was flagged in a previous study[20], highlighting the need for improved constraints that better factor for the composition of coral assemblages within a site.

To better constrain assemblage-specific porosity values for TWA coral reef systems, we analysed imagery of preserved coral assemblages from exposures of well-preserved mid-Holocene and Quaternary interglacial reefs from sites spanning the TWA (Extended Data Fig. 1). Images either were taken by the authorship team or were supplied to us, and derive collectively from locations that would have varied in terms of wave exposure and depositional environment and that include a diversity of preserved coral assemblages. On the basis of a sift for image quality and suitability of more than 90 potentially usable images, 66 images were selected for quantitative analysis of framework porosity. We only selected images taken perpendicular to the exposure face, where sufficient areas of reef framework were exposed and where image quality was sufficient to enable the primary framework-contributing coral components to be clearly differentiated from the surrounding sediment matrix. This approach avoids the potential problems of calculating porosity values from narrow core sequences in which clast movement or poor recovery might complicate interpretations.

On the basis of the corals present in our image datasets, we then attributed each to one of seven distinct coral species–morphotype assemblages representative of deposits formed by the most common TWA shallow-water corals (Supplementary Table 2). Clearly, the abundance of formerly dominant shallow-water branching corals in the region has declined, and many taxa are now present in lower abundances because of coral bleaching, disease and declining water quality[23,50]. As a result, the depth- and wave-energy-influenced patterns of coral zonation that were common in the region before the mid-1970s no longer widely exist[51]. However, the same coral species persist, and therefore the assemblages preserved in these fossil deposits collectively provide an opportunity to quantify the framework porosities generated during clast accumulation by specific assemblages of coral species or morphological groups. Such data are essential if we are to better constrain the links between contemporary coral communities and their contributions to short-term (on the timescale of years to decades) reef accretion potential. This is distinct from longer-term (on the timescale of centuries to millennia) reef-building modes, in which spatial variations in the presence or absence of reef framework (for example, the occurrence of sand channels or framework voids) and the influence of episodic on–off reef movement of framework and/or sediment can create different large-scale framework fabrics.

Each selected image in our dataset was first adjusted for brightness and contrast to enhance the identification of framework components, before importing into Adobe Illustrator (v.2024) (Extended Data Fig. 2). For each image, an area of approximately 0.5 × 0.5 m was delineated to capture the relevant framework components of interest and where the image and exposure quality was sufficiently good. Each selected area thus represents a virtual quadrat for analysis consistent with approaches used for in-situ analyses of palaeo-reef coral assemblages[52]. Within each quadrat, every coral component of the framework was then manually traced and white filled (Extended Data Fig. 2). Only larger framework components (greater than about 5 cm) were delineated, accepting that image quality or coral clast preservation or orientation might have led to some clasts being missed. The key point is that we focused here on what were determined to be primary framework-contributing coral components as opposed to the post-depositional sediment infilling matrix material. The area inside each quadrat was then cut, saved as a .jpg image and exported to Adobe Photoshop (v.2024). Each image was converted to greyscale and the image brightness was adjusted to create a two-tone black-and-white image (Extended Data Fig. 2). This image was then opened in ImageJ (https://imagej.net/ij/) and converted to a binary image, and thresholding analysis was used to calculate the percentage of framework to non-framework components. This provides a measure of the framework stacking porosity space for the image. The images used, image source locations and the resultant porosity values for each class are listed in Supplementary Table 2. To these, we also added an averaged porosity value class of 38.3% as the mean (calculated in Microsoft Excel) of all other class types to apply to highly mixed coral assemblages. It is important here to emphasize that the values we calculate are only from, and therefore most reliably applicable to, TWA reefs. Similar values may be appropriate for equivalent Indo-Pacific coral assemblages, but this ideally needs to be tested in fossil outcrop material from that region.

### Effects of revised porosity values on RAP$_{max}$ rates

To test the effect of using these coral-assemblage-specific porosity values on RAP$_{max}$ rates, we undertook a re-analysis of carbonate production–reef accretion rates previously calculated for a wide diversity of locations across the TWA[3]. We recalculated RAP$_{max}$ for each reef by substituting in the new coral-assemblage-specific porosity factors as appropriate to the types of coral assemblage reported. We used these data to compare RAP$_{max}$ using the original and new values for each reef in each country[3], as well as undertaking an analysis to assess the impact of using the new values across sites as a function

of percentage coral cover and net carbonate production, and for a subset of sites in the Mexican Caribbean to assess the impacts on $RAP_{max}$ rates with different coral-assemblage types. The significance of differences between the old and new values across these analyses was calculated using paired two-sided $t$-tests in GraphPad Prism 10 (Supplementary Table 3).

### Assessing $RAP_{max}$ at regional scales

We then undertook an analysis of the three most comprehensive carbonate budget datasets that now exist for the wider TWA. These are from the Florida Keys, the Mexican Mesoamerican reef and sites in the Gulf of Mexico, and Bonaire. Each provide data with high spatial coverage and high within-subregion replication, thus allowing us to perform a robust assessment of $RAP_{max}$ rates across and within subregions. We focused on reef sites with a depth range of 0.5–6 m (and, for comparative purposes, sites with a depth range of 6–12 m), and for each we first assessed the abundance of each coral species and grouped these by relevant species or coral morphological type in relation to the seven coral-assemblage types delineated in our framework porosity assessment (or, if the range of species present was wide, we used the mixed assemblage class). For each reef, we then assigned an appropriate porosity value based on coral species abundance data, with a proportional contribution of more than 0.6 taken to be indicative of a major contribution for any given species or morphological group (this scoring was undertaken independently by C.T.P. and D.M.d.B., and a final class was agreed), and used these to derive measures of $RAP_{max}$ at each site. In addition, we used regression analysis to investigate the relationship between coral cover and $RAP_{max}$ for each of three common coral-assemblage types: those with *A. palmata*; those dominated by massive corals; and those comprising a mixed assemblage of submassive, encrusting and small-branched coral forms.

### Future projections of $RAP_{max}$ under SLR

To consider the implications of the derived $RAP_{max}$ rates in relation to future projections of SLR and to examine implications for above-reef water depth increases under future warming scenarios, we used published data on the response of coral and coralline algal calcification, and bioerosion from micro- and macro-eroders, to the combined effects of modifications of pH and temperature reported in the literature during laboratory experiments[25]. We did not factor for changing thermal tolerance or for natural or facilitated coral acclimation in these scenarios given ongoing uncertainty about timescales and species-level responses. Linear regression analysis was used to estimate the responses of the factors we consider to the combined effects of both ocean acidification and warming. Projections of proportional changes in coral and coralline algal calcification, and bioerosion from micro- and macro-eroders, were made for each reef following regional SSP projections of temperature and pH changes (Supplementary Table 4). These regional projections used temperature and pH data from the Coupled Model Intercomparison Project 6 (CMIP6; https://esgf-node.llnl.gov/projects/cmip6/) to create a multi-model annual mean for the historical hindcast and each of the four SSP future scenarios (SSP1–2.6, SSP2–4.5, SSP3–7.0 and SSP5–8.5). The multi-model mean used all available climate model outputs to date for temperature (totalling 23, 22, 22, 19 and 23 for historical, SSP1–2.6, SSP2–4.5, SSP3–7.0 and SSP5–8.5, respectively; Supplementary Table 5) and pH (11 for all scenarios; Supplementary Table 6). Models were regridded to a regular grid; data for each site were extracted from the nearest grid point, and the model average for a given site was applied. External bioerosion by fish and urchins was necessarily held as static through our projections owing to an absence of data on future erosion rates by these groups. Our methods deviate from previous methods[25] in that here we follow SSP scenario projections for each reef and replace their estimates of changes in coral cover with those from another report[53]. These determine changes in the annual rate of absolute coral cover based on thermal stress

(using degree-heating week values) with the initial coral cover as a covariate. Here, we apply their approach by calculating the annual maximum degree-heating week values for each site (under each SSP scenario) by summing the positive anomalies above the warmest monthly temperature (30-year baseline between 01/01/1982 and 31/12/2011 for each 12-week period) using the same CMIP6 daily temperature data used for the regional SSP projections. As before, the different climate output models were then combined to create a multi-model mean for each site. We then used this approach to determine percentage reductions in coral cover across three time-period intervals: from date of census to 2040, from 2040 to 2060 and from 2060 to 2100. $RAP_{max}$ rates at the end of each time period were then determined as a function of the percentage change in coral cover and the resulting impact on rates of coral carbonate production, combined with the site-specific impacts of warming and acidification on coral and CCA calcification and micro- and macro-endolithic erosion. We then compared the resulting projections of $RAP_{max}$ for each time point (2040, 2060 and 2100) and under each SSP scenario against rates of SLR (Supplementary Table 7), using SLR rate projections based on the findings of Chapter 9 of the Working Group 1 contribution to the IPCC Sixth Assessment Report[28] and the Framework for Assessment of Changes To Sea-level (FACTS)[54], accessed using the NASA Sea Level Projection Tool[55]. We acknowledge that magnitudes of SLR under each SSP scenario might change as projections are further refined.

### Reporting summary

Further information on research design is available in the Nature Portfolio Reporting Summary linked to this article.

## Data availability

The supplementary files include details of the reef framework imagery analysed; SSP-aligned rates for determining coral cover, coral calcification, CCA calcification and bioerosion changes; climate models used for SST and pH projections; and SSP-aligned data on SLR by subregion. Additional site-specific rate data supporting this publication are openly available from the University of Exeter's institutional repository at https://doi.org/10.24378/exe.5766. We acknowledge the IPCC AR6 Sea Level Projection Tool web page (https://toolkit.climate.gov/tool/ipcc-ar6-sea-level-projection-tool).

## Code availability

Computer code used to produce the projections of coral cover change, and the effects of SST and ocean acidification on coral and coralline calcification and on substrate bioerosion rates, is freely available at https://github.com/ComeauS/Perry_et_al_Caribbean/tree/main.

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

**Acknowledgements** C.T.P. and D.M.d.B. were funded under grant RPG-2021-295 from the Leverhulme Trust. B.H. was supported by Japan Society for the Promotion of Science (JSPS)

KAKENHI grant 23K26924. C.E.C was supported under the Coastal People Southern Skies Centre of Research Excellence, New Zealand. We thank K. Johnson, D. Muhs and P. Blanchon for providing additional framework imagery. Any use of trade, firm, or product names is for descriptive purposes only and does not imply endorsement by the US government. For the purpose of Open Access, the author has applied a CC BY public copyright licence to any Author Accepted Manuscript version arising from this submission.

**Author contributions** C.T.P. conceived the project and led the analysis and writing, with major inputs from D.M.d.B. and A.E.W. Palaeo-imagery and contextual data were provided by C.T.P., A.O., E.M.D., L.T.T., E.H.M. and W.F.P. C.T.P. and D.M.d.B. analysed the imagery. D.M.d.B., E.H.M., L.A.-F., E.P.-C., J.M. and I.C.E. provided carbonate budget datasets. C.E.C., S.C. and B.P.H. provided analyses of future coral cover and calcification and bioerosion rates. All authors made substantive contributions to the text. The scientific results and conclusions, as well as any views or opinions expressed herein, are those of the authors and do not necessarily reflect those of Ocean and Atmospheric Research at NOAA or the Department of Commerce.

**Competing interests** The authors declare no competing interests.

**Additional information**
**Correspondence and requests for materials** should be addressed to Chris T. Perry.

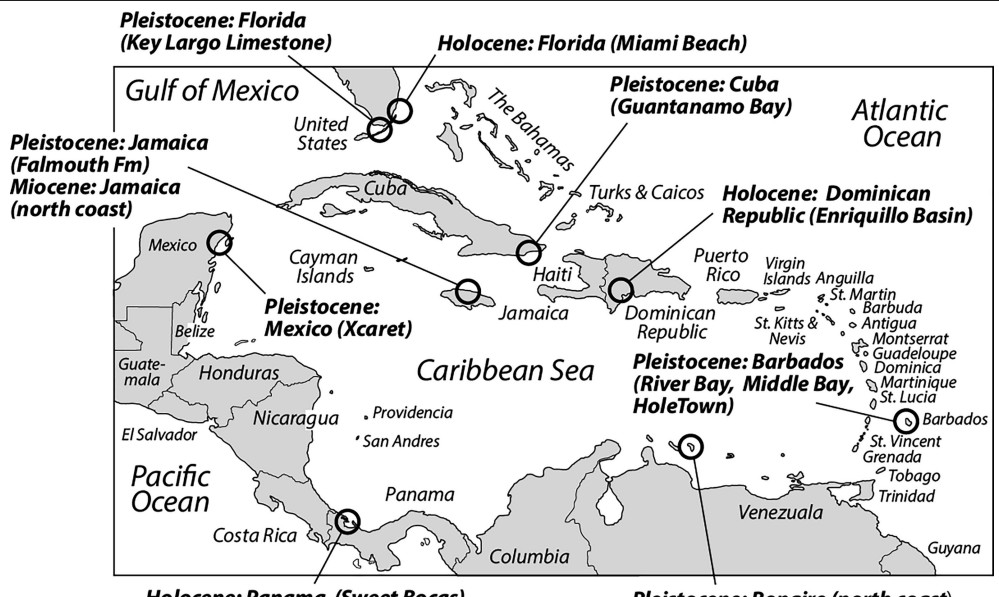

**Extended Data Fig. 1 | Locations of reef deposits analysed to assess framework porosity values.** Map showing the location of Holocene and Pleistocene reef deposits across the western Atlantic region from which fossil coral assemblage imagery was used for framework analysis.

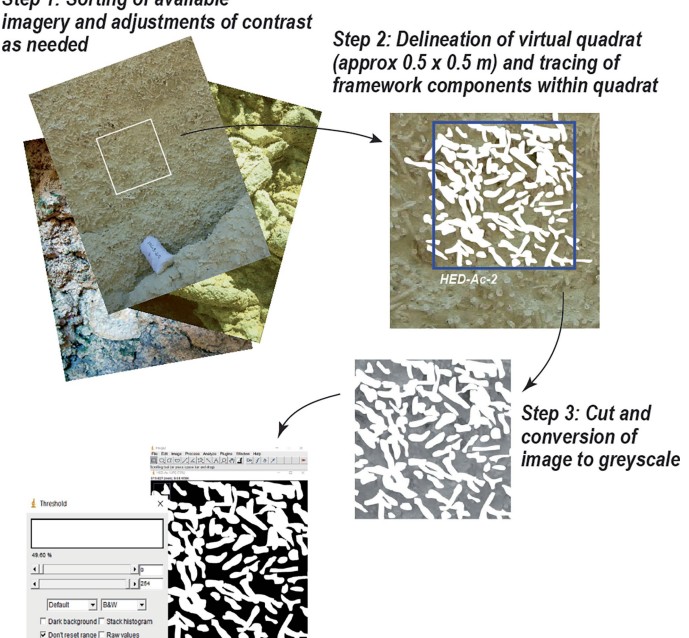

*Step 1: Sorting of available imagery and adjustments of contrast as needed*

*Step 2: Delineation of virtual quadrat (approx 0.5 x 0.5 m) and tracing of framework components within quadrat*

HED-Ac-2

*Step 3: Cut and conversion of image to greyscale*

*Step 4: Conversion to binary image and thresholding to determine % framework components*

**Extended Data Fig. 2 | Steps in the reef deposit image sorting and analysis procedure.** Sequences of steps involved in the selection, tracing and thresholding of reef outcrop imagery to determine the ratio of primary framework to non-framework components, and from this, stacking porosity values. Example shown from an *A. cervicornis*-dominated sequence from the Holocene of the Enriquillo Basin, Dominican Republic.

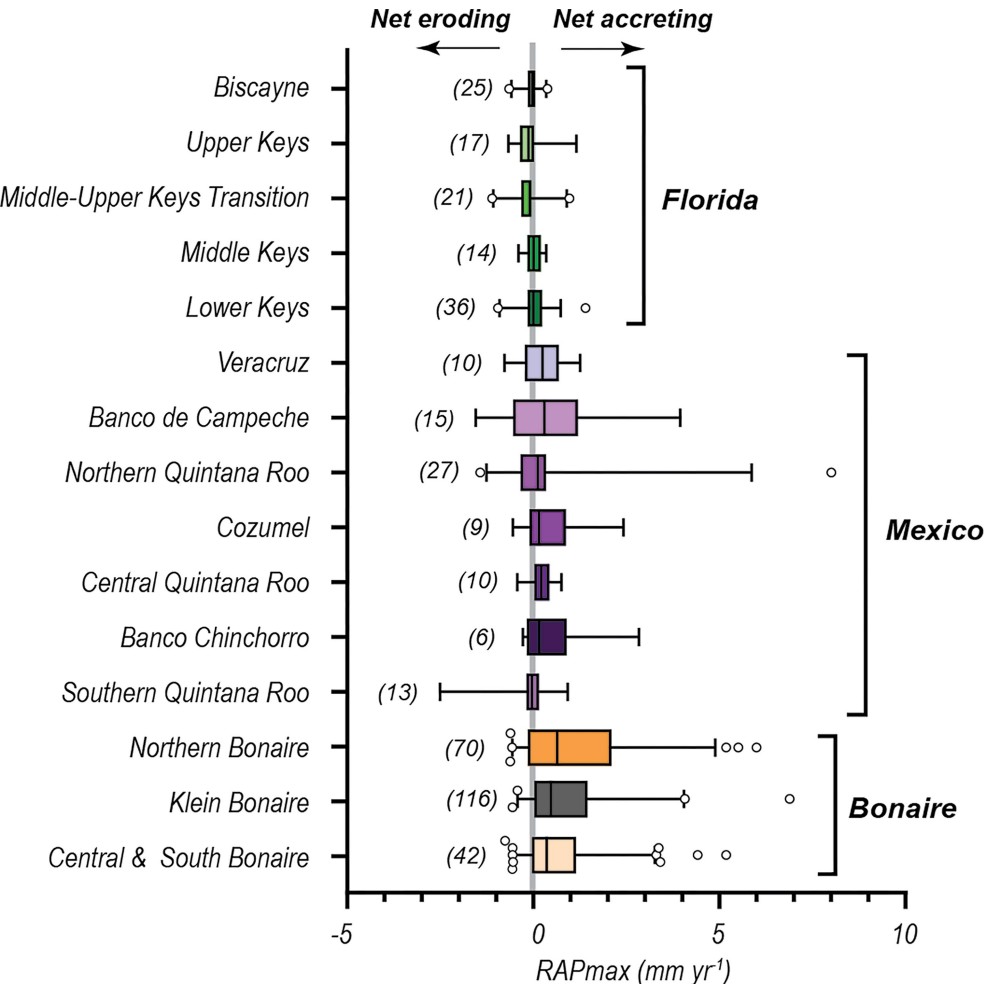

**Extended Data Fig. 3 | Calculated contemporary western Atlantic RAP$_{max}$ rates (mm yr$^{-1}$).** Calculated contemporary RAP$_{max}$ rates (mm yr$^{-1}$) for each subregion within each country. Italics in parentheses are number of replicates per location. Box plots depict median (vertical line within box), box widths depict first and third quartiles, whiskers represent the 95th percentile. Outliers outside the 95th percentile are shown as circles.

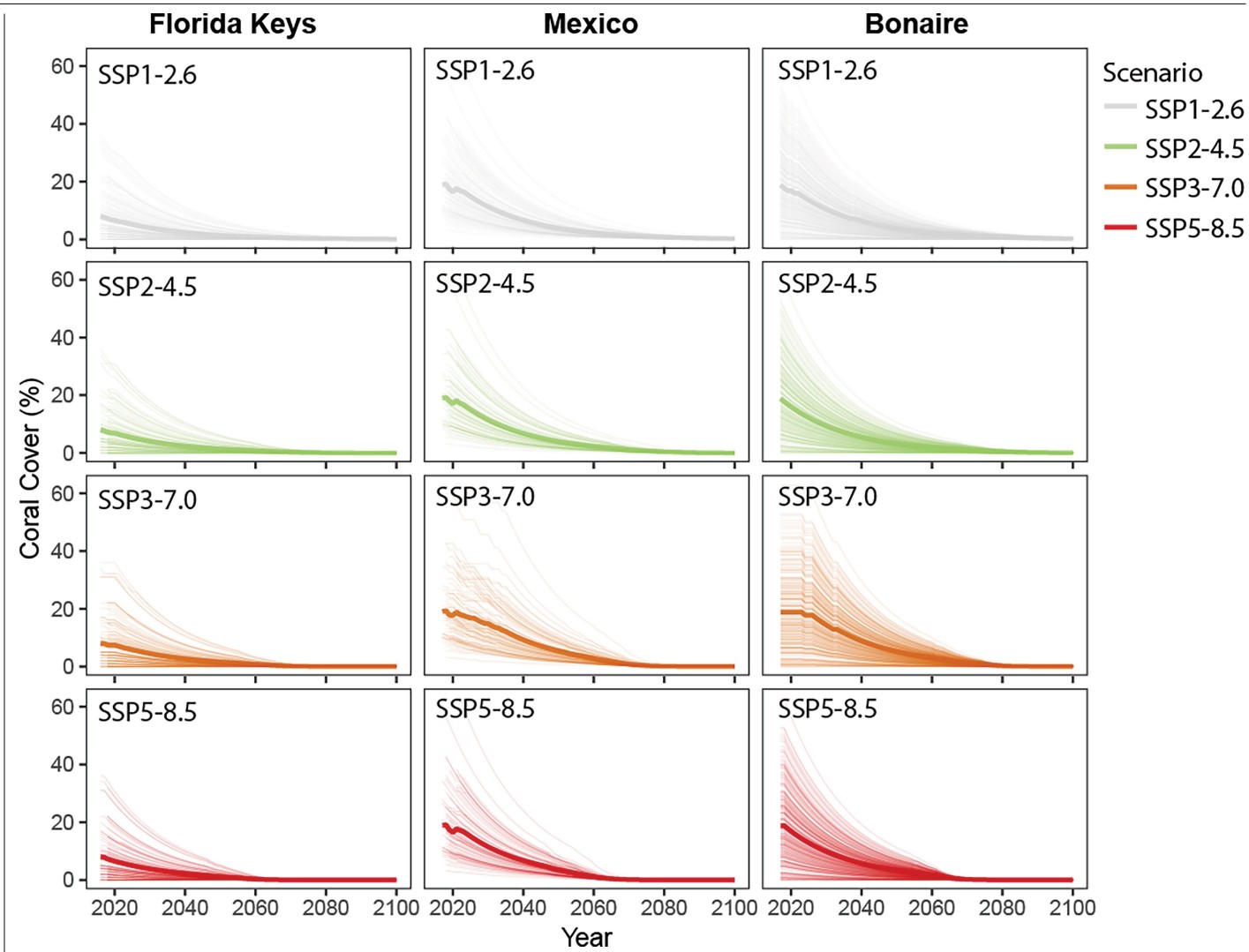

**Extended Data Fig. 4 | Modelled trends in western Atlantic coral reef cover.** Modelled trends in coral cover for each subregion and under each SSP scenario through to 2100.

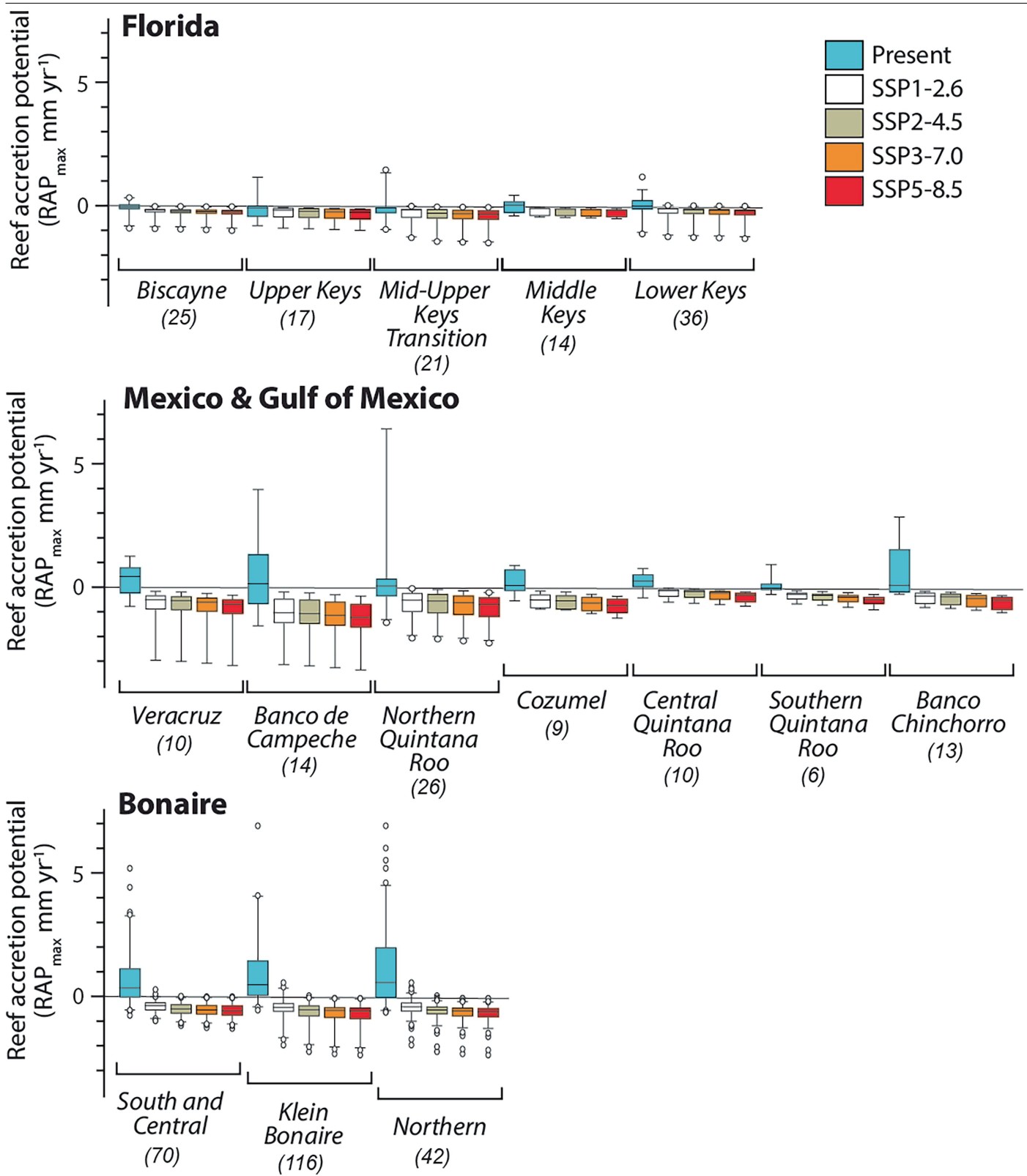

**Extended Data Fig. 5 | Projected future changes in RAP$_{max}$ rates under various SSP scenarios.** RAP$_{max}$ rates (mm yr$^{-1}$) at present compared to 2100 under different SSP scenarios (SSP1–2.6; SSP2–4.5, SSP3–7.0 and SSP5–8.5) in each subregion within each country. Italics in parentheses are number of replicates per scenario. Box plots depict median (horizontal line), box height depicts first and third quartiles, whiskers represent the 95th percentile. Sites outside the 95th percentile are shown as circles.

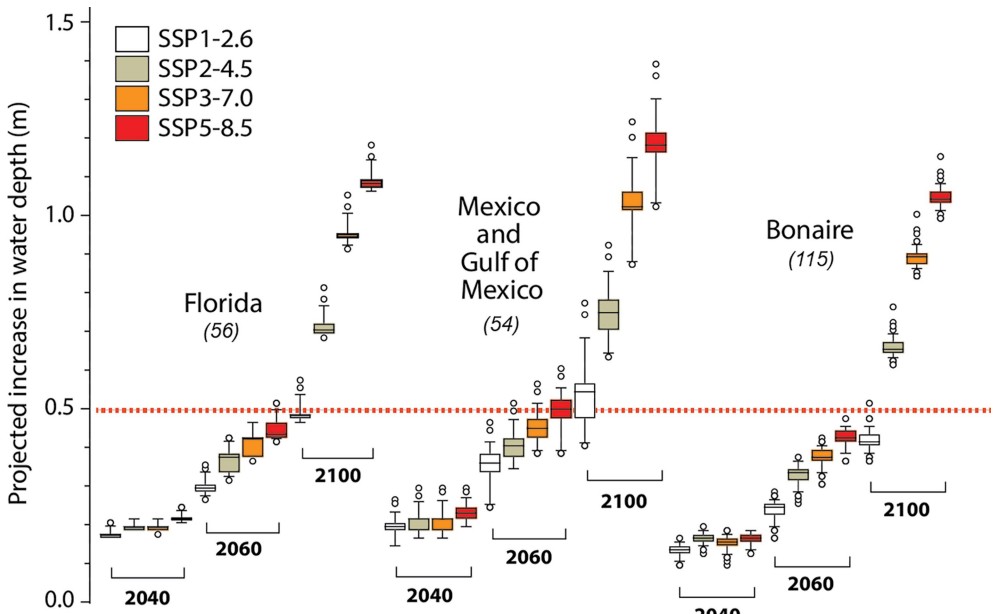

**Extended Data Fig. 6 | Projected increases in total water depth above fore-reef zones.** Projected increases in total water depth (m) above fore-reef (6–12 m depth) zones of reefs in each country under each SSP scenario at 2040, 2060 and 2100. Italics in parentheses are number of replicates per scenario and time point. Box plots depict median (horizontal line), box height depicts first and third quartiles. Sites outside the 95th percentile are shown as circles. White circles show outliers. Dashed horizontal line represents the 0.5 m water depth increase point.

# Reporting Summary

## Statistics

For all statistical analyses, confirm that the following items are present in the figure legend, table legend, main text, or Methods section.

| n/a | Confirmed | |
|---|---|---|
| ☐ | ☒ | The exact sample size (*n*) for each experimental group/condition, given as a discrete number and unit of measurement |
| ☐ | ☒ | A statement on whether measurements were taken from distinct samples or whether the same sample was measured repeatedly |
| ☐ | ☒ | The statistical test(s) used AND whether they are one- or two-sided<br>*Only common tests should be described solely by name; describe more complex techniques in the Methods section.* |
| ☐ | ☒ | A description of all covariates tested |
| ☐ | ☒ | A description of any assumptions or corrections, such as tests of normality and adjustment for multiple comparisons |
| ☐ | ☒ | A full description of the statistical parameters including central tendency (e.g. means) or other basic estimates (e.g. regression coefficient) AND variation (e.g. standard deviation) or associated estimates of uncertainty (e.g. confidence intervals) |
| ☐ | ☒ | For null hypothesis testing, the test statistic (e.g. *F*, *t*, *r*) with confidence intervals, effect sizes, degrees of freedom and *P* value noted<br>*Give P values as exact values whenever suitable.* |
| ☐ | ☒ | For Bayesian analysis, information on the choice of priors and Markov chain Monte Carlo settings |
| ☐ | ☒ | For hierarchical and complex designs, identification of the appropriate level for tests and full reporting of outcomes |
| ☐ | ☒ | Estimates of effect sizes (e.g. Cohen's *d*, Pearson's *r*), indicating how they were calculated |

*Our web collection on statistics for biologists contains articles on many of the points above.*

## Software and code

Policy information about availability of computer code

| | |
|---|---|
| Data collection | Fossil imagery data used to assess reef framework stacking porosity values was compiled from existing collections from across the authorship team. Carbonate budget data reported here were collected using the established ReefBudget protocol (www.exeter.ac.uk/research/projects/geography/reefbudget/) by regional members of the authorship team between 2016 and 2023. Data for comparative analysis to assess the impact of using the new reef stacking porosity values on reef accretion rates was undertaken against data used in a 2018 Nature paper (doi.org/10.1038/s41586-018-0194-z). |
| Data analysis | Fossil reef imagery data were analysed by importing imagery first into Adobe Illustrator (version 2024) to delineate framework components, then into Adobe Photoshop (version 2024) for conversion to B&W images, and then into ImageJ (Dec 2023 version) to determine the % of framework to non-framework components using thresholding analysis. Analysis of contemporary reef carbonate budget data and of estimated reef accretion potential rates, both today and under future SST and OA effects, was undertaken using the following: R (V4.0.4), CDO (1.9.10), Image J (Dec 2023 version), Microsoft Excel (2024), and GraphPad Prism 10. Computer codes used to produce the projections of coral cover change and SST and OA impacts on coral and coralline calcification, and on substrate bioerosion rates are freely available at https://github.com/ComeauS/Perry_et_al_Caribbean/tree/main. The following packages were used in R: Tidyverse #Data processing and organisation (v2.0.0); Future #For Parallel processing (v1.33.1); Tidync #For reading netcdf files (v0.3.0); Data.table #Data processing and organisation (v1.16.4); Arrow #Write compressed output files (v9.0.0.20); sf #spatial analysis (v1.0-15); Plyr (v1.8.9). Projected rates of reef accretion were compared to projected rates of sea-level rise using projections based on the findings of Chapter 9 of the Working Group 1 contribution to the IPCC Sixth Assessment Report, the Framework for Assessment of Changes To Sea-level (FACTS) and accessed via the NASA Sea Level Projection Tool (https://toolkit.climate.gov/tool/ipcc-ar6-sea-level-projection-tool). |

For manuscripts utilizing custom algorithms or software that are central to the research but not yet described in published literature, software must be made available to editors and reviewers. We strongly encourage code deposition in a community repository (e.g. GitHub). See the Nature Portfolio guidelines for submitting code & software for further information.

## Data

Policy information about availability of data

All manuscripts must include a data availability statement. This statement should provide the following information, where applicable:
- Accession codes, unique identifiers, or web links for publicly available datasets
- A description of any restrictions on data availability
- For clinical datasets or third party data, please ensure that the statement adheres to our policy

Supplementary information and supporting data is provided as a pdf file. Additional site-specific rate data supporting this publication are openly available from the University of Exeter's institutional repository at: https://doi.org/10.24378/exe.5766.

## Research involving human participants, their data, or biological material

Policy information about studies with human participants or human data. See also policy information about sex, gender (identity/presentation), and sexual orientation and race, ethnicity and racism.

| | |
|---|---|
| Reporting on sex and gender | No human participants or their data or biological material were used in this study. |
| Reporting on race, ethnicity, or other socially relevant groupings | No human participants or their data or biological material were used in this study. |
| Population characteristics | No human participants or their data or biological material were used in this study. |
| Recruitment | No human participants or their data or biological material were used in this study. |
| Ethics oversight | No human participants or their data or biological material were used in this study. |

Note that full information on the approval of the study protocol must also be provided in the manuscript.

# Field-specific reporting

Please select the one below that is the best fit for your research. If you are not sure, read the appropriate sections before making your selection.

☐ Life sciences ☐ Behavioural & social sciences ☒ Ecological, evolutionary & environmental sciences

For a reference copy of the document with all sections, see nature.com/documents/nr-reporting-summary-flat.pdf

# Ecological, evolutionary & environmental sciences study design

All studies must disclose on these points even when the disclosure is negative.

| | |
|---|---|
| Study description | This study explores future coral reef growth potential under projected rates of sea-level rise. Specifically, it constrains time points and magnitudes of water depth increases above coral reefs under different Shared Socioeconomic Pathway (SSP) emission scenarios through to 2100. |
| Research sample | Data on rates of reef carbonate production and erosion were used to describe the carbonate budget state of reefs at discrete reef sites located along the Florida Keys, in the Gulf of Mexico and along the Caribbean coast of Mexico, and from around the island of Bonaire. These data were previously collected by members of the authorship team and provide the basis for the assessments of reef accretion potential through to 2100 we conduct here. Underpinning the reef accretion assessments are data on reef framework stacking porosity values calculated from images collected by the authorship team from fossil reef outcrops in the Caribbean. |
| Sampling strategy | Carbonate budget data used in this study were previously collected by in-country teams from sites spanning a range of water depths (0.5-12 m depth) at predetermined sampling locations in each region of interest. The central aim of this data collection was to capture data from a range of sites and water depths in each location. |
| Data collection | All fossil reef imagery used in the analysis of reef framework stacking porosity values was collected by members of the authorship team or as described in SI Table 2. Carbonate budget data used in the main analysis reported here were collected using the established ReefBudget protocol (www.exeter.ac.uk/research/projects/geography/reefbudget/) by regional members of the authorship team between 2016 and 2023. Comparative data analysis to assess the impact of using the new reef stacking porosity values was undertaken against data used in a 2018 Nature paper (doi.org/10.1038/s41586-018-0194-z). |
| Timing and spatial scale | Underpinning ReefBudget data used in this study as baseline start points for our analysis were collected at different time periods in each location as a function of the timing of the specific projects involved in each region: Florida - 2016 and 2018; Mexico - between 2017 and 2022; Bonaire - 2017. |

| | |
|---|---|
| Data exclusions | The only data excluded from our analysis were any data collected from sites >12 m depth as we deemed these to be not relevant to determining shallow water reef accretion behaviour. |
| Reproducibility | Data collection sites and water depths are listed in SI Table 8 for each country. |
| Randomization | ReefBudget data were grouped for each country into geographic sub-regions in each country and then additionally sorted by depth category (0-6m and 6-12 m) |
| Blinding | Not applicable as data analysis was based on in-field surveys along transect lines. |

Did the study involve field work?  ☒ Yes  ☐ No

## Field work, collection and transport

| | |
|---|---|
| Field conditions | As typical with all marine based in-water surveying, in-water conditions were variable but any day to day variations in conditions are considered unlikely to have had a major influence on the metrics collected along the benthic transects. |
| Location | In-field survey data were collected from individual reef sites/water depths at locations spanning the three main regions of interest in Florida, Mexico and Bonaire. Data from each site in provided in SI Table 8. |
| Access & import/export | No import/export issues as all data recorded in-field. |
| Disturbance | Data collection and surveying used temporary placed transect lines, removed after each survey, and were non-invasive. |

# Reporting for specific materials, systems and methods

We require information from authors about some types of materials, experimental systems and methods used in many studies. Here, indicate whether each material, system or method listed is relevant to your study. If you are not sure if a list item applies to your research, read the appropriate section before selecting a response.

## Materials & experimental systems

| n/a | Involved in the study |
|---|---|
| ☒ ☐ | Antibodies |
| ☒ ☐ | Eukaryotic cell lines |
| ☒ ☐ | Palaeontology and archaeology |
| ☒ ☐ | Animals and other organisms |
| ☒ ☐ | Clinical data |
| ☒ ☐ | Dual use research of concern |
| ☒ ☐ | Plants |

## Methods

| n/a | Involved in the study |
|---|---|
| ☒ ☐ | ChIP-seq |
| ☒ ☐ | Flow cytometry |
| ☒ ☐ | MRI-based neuroimaging |

## Plants

| | |
|---|---|
| Seed stocks | Not used |
| Novel plant genotypes | Not used |
| Authentication | Not used |

