## [Peer Review File · Nature]

Reduced Atlantic reef growth past 2°C warming amplifies sea-level impacts

Corresponding Author: Professor Chris Perry

Version 0:

Reviewer comments:

Referee #1

(Remarks to the Author)

In this study, Chris Perry et al. follow up on their previous research published in Nature seven years ago. This time, they focus exclusively on Tropical Western Atlantic reefs, whereas their earlier study included both Tropical Western Atlantic and Indian Ocean reefs. The key breakthrough in this study is the incorporation of fossil reef deposit analysis, which revises their previous porosity estimates to an average of 38%—significantly lower than earlier values for branch-dominated reefs (80%) and mixed-dominated reefs (50%).

Overall, this study is well-conducted, well-written, and will undoubtedly be groundbreaking for the coral reef science community. However, I found one major aspect of the conclusions frustrating: the assertion that water depth above reefs will increase, without considering the potential for these reefs to expand vertically needs to be discussed. If water depth increases above a reef, similar hydrodynamic conditions to those currently experienced by corals may occur at shallower depths. Given that hydrodynamics play a key role in vertical coral distribution (see Madin & Connolly, 2006; Madin et al., 2014), this possibility should at least be discussed. Or at least, the fact they consider that corals are not shifting to shallower water can be discussed in the face of thermal anomalies, which may instead confine coral reefs to their current lower depths (see Jorda et al., 2020).

The authors conclude their manuscript with a modeling effort on reef restoration, which is both valuable and interesting. However, it appears to focus on the same coral zones—if this is not the case, it should be clarified in the methods. While this is not a critical flaw, it should be acknowledged that projections of water depth apply to current reef zones. A few lines addressing this in the discussion would strengthen the manuscript. I would also strongly recommend reflecting this nuance in the title, as it could shift the study's tone from highly pessimistic to one that offers a glimmer of hope.

Specific comments:

- Line 31: The function itself is not threatened, but it may be reduced.
- Figure 1A: It is unclear how you defined the core average porosity. This is not explained in the caption, the main text, or the methods. Looking at this range, I wonder whether porosity has changed over time. Is it possible that porosity was higher in the past but has decreased due to global change? My hypothesis is that while corals continue to grow, their structural quality may have declined over time.
- Lines 253-260: You mention that water levels above many reefs would double, but according to SI Table 8, the minimum depth is 3. Could you clarify what you mean? This ties back to my general comment about the potential for reefs to expand their vertical range.
- Lines 297-302: These results suggest that branched corals are the preferred reef-dominant type, as reefs with 40% branched coral cover display a minimum RAPmax, whereas massive species require 70% cover to achieve the same effect. This conclusion aligns with the structural complexity values you reported in Husband et al. (2021), which showed higher complexity for branched corals. Since the three key variables influencing flood risk reduction are RAPmax, SLR, and structural complexity, I encourage you to elaborate some lines on the role of structural complexity. You mention it only briefly in the concluding section, but previous studies have demonstrated its equal importance in mitigating extreme wave run-up (see Carlot et al., 2023) and wave increase (see Harris et al., 2018).
- Lines 541-542: Was this done manually, or did you use AI assistance? Given the substantial workload involved, it would be worth specifying whether the process was manual or supported by an AI-enhanced workflow.

• Lines 576-581: I do not fully follow the rationale here. You defined porosity based on seven assemblages (Figure 1), and you also have coral cover data as well as assemblage information. Why did you not use a weighted porosity approach, defined as:

Reef porosity(i) = $\sum (\text{cover}(i) \times \text{Porosity}(i))$; with the cover expressed from 0 to 1 instead of having percentage?

Jérémy Carlot

Post-doctoral researcher at IEO, Spain

Referee #2

(Remarks to the Author)

The manuscript entitled 'Substantial water depth increases above coral reefs unavoidable beyond 2oC warming' provides new and valuable insights into the potential for coral reefs in the Tropical Western Atlantic to keep up with future sea level rises. Overall, I thoroughly enjoyed reading the paper, it was well written with excellent figures and well referenced. I do have three areas where some more clarity or inclusion of discussion and/or detail might benefit the paper. These relate to: 1) reef framework porosity and sediment infilling, 2) future projections of carbonate budgets (and therefore related RAP calculations, and 3) accommodation space over reef flats (see below for further details). Regardless, the paper will be of broad interest and highlights the need for out of the box ideas around how we can upscale reef restoration efforts that increase a reefs carbonate budget and accretionary capacity. Hence, I recommend the paper for publication.

One of the key aspects that I applaud the authors for tackling is improvements to how reef accretionary potential (RAP) has been calculated in the past, which has involved a guesstimate of reef porosity. Here the authors, through the use of the fossil record, have devised a new and more quantifiable approach to assessing reef porosity – here termed stacking porosity. Stacking porosity varies between reefs due to variations in coral assemblages. My question here to the authors is that this technique would be somewhat limited to clearwater reefs given that we also know (albeit not very well quantified) of the importance of sediment infilling which can also 'enhance' reef accretionary rates by filling in the pores in the accreting reef structure. Hence there is potential here that these calculated porosity values which are from the fossil record – miss this important element that can facilitate reef accretion, and thereby underestimate accretionary potential.

My second question relates to future predictions of net carbonate production, which the authors have calculated using future projections of temperature and pH models. I understand why the authors have used this approach, but looking over table S4 – declines in coral cover, calcification rates etc are seemingly generic (i.e. the same over time), and if there is one thing I've learnt from working on coral reefs is of the highly heterogenous nature within let alone between reefs. So to suggest that all reefs will respond to the same degree I think is a huge simplification of how reefs will respond to these future stressors. I also understand that getting these types of site specific data is hard – but I would like to see some acknowledgement that not all reefs are the same for both internal and external reasons. Calculations of carbonate budgets involve a lot of variability along the way, which is often averaged, then averaged, then averaged again – so the true error of these calculations is not always transparent. So I would just be mindful of this.

Finally, I always wonder – are there reefs (which may not include these reefs here as I am less familiar with Caribbean reef systems) – where an increase in sea level be a good thing for carbonate production? That is, for those reef flats (which can be vast areas in the 10 km²), which are essentially at a hiatus and corals living in these environments are relatively hard core i.e. they are dealing with extreme conditions and yet surviving. Could an increase in SL allow these corals to grow more, there by increasing carbonate production and therefore reef accretionary potential. Food for thought. Perhaps not easily woven in here, or perhaps something to add in as a discussion point???

Nicola Browne

Referee #3

(Remarks to the Author)

The manuscript by Perry and collaborators presents a valuable assessment of the impacts of sea-level rise (SLR) on Caribbean coral reef environments and highlights the importance of proactive measures to mitigate some of the negative effects. Notably, the modelling approach includes optimistic scenarios in which strong and aggressive restoration efforts are implemented. The study offers a novel perspective by integrating a reassessment of geological data with contemporary coral growth and cover information, along with climate scenario modelling. Overall, the manuscript is well written, the topic is timely and relevant, and the analysis is robust. I therefore support its publication, pending consideration of a few minor comments.

One of the central findings appears to be the projected near-total collapse of reef accretion by 2040 across all scenarios. To what extent has the potential for coral acclimatization, e.g. genetic selection, or modulation by climatic modes of variability been considered in these projections? Line 339 mentions that acclimatization/adaptation could occur — has this been incorporated into the model, or is it acknowledged as an uncertainty?

Additionally, has the changes in physical oceanography associated to increased water depth been considered in the models? For instance, might it enhance water exchange with cooler oceanic waters or facilitate the removal of nutrients and pollutants, thereby mitigating some stressors? Conversely, could it worsen conditions by increasing water column stratification and reducing vertical mixing?

Line 74 – This statement is somewhat unclear. Since SLR is itself a consequence of climate change, it would be helpful to clarify here that you are referring to the direct (non-SLR) impacts of climate change.

Figures 1 and 2 – RAP is defined as reef accretion rates rather than reef accretion potential. Please clarify the terminology to ensure consistency and prevent confusion.

Version 1:

Reviewer comments:

Referee #1

(Remarks to the Author)

After carefully reading the revised manuscript and the authors' point-by-point responses, I am satisfied that all my concerns have been thoroughly addressed. The authors have provided clear clarifications and made meaningful improvements to the manuscript. This revised version represents a strong and valuable contribution to the field, and in my view, it meets the standards for publication in Nature.

Jérémy Carlot

Referee #2

(Remarks to the Author)

The authors have satisfactorily meant my previous comments and (as far as I can tell) the comments from the other two reviewers. I once again commend the authors on a job well done. I thoroughly enjoyed the paper, was excited to see a new and improved method for assessing porosity and the re-evaluation of the RAP calculations. I think this paper will be of broad interest and be highly cited due to its new methodological approach incorporated into the paper. Hence recommendation to publish as is.

Referee #3

(Remarks to the Author)

The authors have satisfactorily addressed all the points raised during the review process, and the revised manuscript is substantially improved. I support the publication of the manuscript in its current form. While I acknowledge that some aspects discussed in the review are difficult to incorporate directly into the models, the authors' recognition of these limitations and the associated uncertainties is appropriate and important.

2025-01-01768

Revised title to fit character limit: Reduced Atlantic reef growth past 2°C warming amplifies sea-level impacts

We thank the reviewers for their time and care in reviewing our submission and were delighted to read the three very positive sets of comments. Below we provide our responses and outline how and where we have responded to their comments.

Referees' comments:

Responses below – *line numbers referred to relate to those in the track changed version*

Referee #1 (Remarks to the Author):

In this study, Chris Perry et al. follow up on their previous research published in Nature seven years ago. This time, they focus exclusively on Tropical Western Atlantic reefs, whereas their earlier study included both Tropical Western Atlantic and Indian Ocean reefs. The key breakthrough in this study is the incorporation of fossil reef deposit analysis, which revises their previous porosity estimates to an average of 38%—significantly lower than earlier values for branch-dominated reefs (80%) and mixed-dominated reefs (50%).

Comment 1.1. Overall, this study is well-conducted, well-written, and will undoubtedly be groundbreaking for the coral reef science community. However, I found one major aspect of the conclusions frustrating: the assertion that water depth above reefs will increase, without considering the potential for these reefs to expand vertically needs to be discussed. If water depth increases above a reef, similar hydrodynamic conditions to those currently experienced by corals may occur at shallower depths. Given that hydrodynamics play a key role in vertical coral distribution (see Madin & Connolly, 2006; Madin et al., 2014), this possibility should at least be discussed. Or at least, the fact they consider that corals are not shifting to shallower water can be discussed in the face of thermal anomalies, which may instead confine coral reefs to their current lower depths (see Jorda et al., 2020).

*Response: We understand this point and it is a valid one – especially in an Indo-Pacific context (more on this below). However, it is also important to emphasise that the geomorphic structure of Western Atlantic reefs (which have very narrow crest/back-reef zones and which are mostly not yet sea-level constrained) is very different to that across many Indo-Pacific (and especially Pacific) reefs which commonly exhibit expansive reef flats that are low tide emergent. Large reef flats of this type do not occur in the TWA. This fact combined with the 1) current very low coral cover states (mostly <15%) and 2) projected speed of further coral cover losses, means that neither a physical location on the reef i.e., a reef flat that could be opened up to recolonisation by sea level rise (SLR) nor a coral community to drive that renewed vertical reef growth, makes this reef shallowing scenario likely for the TWA. The point of the reviewer is clearly a valid one in the Indo-Pacific. The reef flats in that region could, as suggested, have new coral growing on them following SLR – or at least there is the potential for that to occur. This will depend on future coral persistence on these reefs, but we agree the potential exists. In light of this we feel it is appropriate to add a brief comment to the M/S emphasising that response potential may be different in the Pacific - references to support this are also added. **See Line 249 onwards.** We have in addition (partly also linked to Reviewer 3 comment) added a note on the potential for changes to nearshore hydrodynamics and ecologies and reef-open ocean water exchange (**Line 50 and 240**).*

Comment 1.2. The authors conclude their manuscript with a modeling effort on reef restoration, which is both valuable and interesting. However, it appears to focus on the same coral zones—if this is not the case, it should be clarified in the methods. While this is not a critical flaw, it should be acknowledged that projections of water depth apply to current reef zones. A few lines addressing this in the discussion would strengthen the manuscript.

*Response: Again, this is a valid point but actually for the same reasons given above, restoration of Caribbean reefs has limited vertical (shallower zone) planting potential. The only habitat to plant corals in with any relevance to improving wave attenuation through vertical growth or enhanced structural complexity are the shallow (and narrow) crest/back-reef zones that define the regions reefs. Geomorphologically SLR will not open up new habitat areas where restoration could be applied, so out-planting can only occur in the same upper reef depth zones that presently exist. Again, we comment briefly on this. **Line 264.***

Comment 1.3. I would also strongly recommend reflecting this nuance in the title, as it could shift the study's tone from highly pessimistic to one that offers a glimmer of hope.

Response: In light of this comment, and we agree nuance is important, we have removed "unavoidable" from the title. The title in any case needs shortening to fit Nature requirements (75 characters with spaces) and alongside the removal of unavoidable we have added specific reference to the Atlantic – this should avoid any unintended confusion about the theoretical possibility of different responses in the Pacific.

Specific comments:

- Line 31: The function itself is not threatened, but it may be reduced.

Wording modified in the abstract

- Figure 1A: It is unclear how you defined the core average porosity. This is not explained in the caption, the main text, or the methods. Looking at this range, I wonder whether porosity has changed over time. Is it possible that porosity was higher in the past but has decreased due to global change? My hypothesis is that while corals continue to grow, their structural quality may have declined over time.

*Porosity here, or more specifically stacking porosity, as used in the original budget to accretion conversion approach refers not directly to skeletal density/porosity, but to the larger scale porosity associated with the accumulation of coral framework to non-coral framework elements (sediment, void spaces) of a reef deposit. This is defined in **lines 96** and also in the Methods (**L510 onwards**). This stacking porosity is controlled by the types of corals present in the community and which are then deposited on the reef – and thus it is their growth morphology and the way these morphologies breakdown post-mortem that is the main influence on stacking porosity (**please see Line 512 onwards for a discussion**). As requested on **L583** we also now make clear how the average porosity value was derived. The skeletal porosity/density that we think the reviewer is referring to – apologies if we have misunderstood the comment - may well be changing under changing SST and OA conditions (and there is some evidence of this published). However, the central point here is that we are not aware of any work showing that this is changing the overall morphologies of corals nor the way skeletons breakdown post-mortem. Indeed, the depositional nature of the fossil deposits we have analysed are very similar to those we can observe accumulating on the present surfaces of reefs where a 3-D sense of those modern deposits can be discerned.*

- Lines 253-260: You mention that water levels above many reefs would double, but according to SI Table 8, the minimum depth is 3. Could you clarify what you mean? This ties back to my general comment about the potential for reefs to expand their vertical range.

*The minimum depth of our study reefs is not 3 m but 1m in several locations (see sites lower down in SI Table 8) – about the minimum workable depth for surveying. This is close to the shallowest zones (i.e., the uppermost reef surfaces depth-wise which in many TWA locations generally lie at about 0.5 m below mean water level). In the text we note two things – 1) that “most reefs will progressively experience more than 0.5 m of water depth increase” – what we are saying here is that water depths will be increasing by ~0.5 m minimum relative to the present reef surface regardless of current depth profile of a reef (see **addition line 222 on**) – because there is no evidence that the upper ~10 m of reefs will grow, and 2) “This would approximately double mean water levels above the shallowest areas of many TWA reefs” i.e., the net effect of SLR and reduced reef growth will on average be a doubling of water levels over the shallowest parts of TWA reefs (as we comment above the shallow crest/back-reef zones that define many TWA reefs are at about 0.5 m depth at mean water). A 0.5 m net increase doubles this. This is a very different situation to Pacific reefs where of course many reef flats are exposed at low water.*

- Lines 297-302: These results suggest that branched corals are the preferred reef-dominant type, as reefs with 40% branched coral cover display a minimum RAPmax, whereas massive species require 70% cover to achieve the same effect. This conclusion aligns with the structural complexity values you reported in Husband et al. (2021), which showed higher complexity for branched corals. Since the three key variables influencing flood risk reduction are RAPmax, SLR, and structural complexity, I encourage you to elaborate some lines on the role of structural complexity. You mention it only briefly in the concluding section, but previous studies have demonstrated its equal importance in mitigating extreme wave run-up (see Carlot et al., 2023) and wave increase (see Harris et al., 2018).

*The reviewer is correct that higher structurally complex corals, which are also the fastest growing, will also help with wave attenuation for frictional reasons. Indeed, we do say this (**lines 296**) “Given the current state of many TWA reefs, any actions that generate additional framework-building are helpful, not least because reef-derived coastal protection benefits are additionally supported by enhanced physical structural complexity (Harris et al. 2018). We are very tight on word space here but as requested have very briefly extended these points about structural complexity gains and wave attenuation benefits. **Line 299**.”*

- Lines 541-542: Was this done manually, or did you use AI assistance? Given the substantial workload involved, it would be worth specifying whether the process was manual or supported by an AI-enhanced workflow.

*This analysis was done manually (now made clear in the Methods – **line 574**). We initially explored AI workflow options but image and clast exposure heterogeneity, different image coloration, partial shading issues etc made the volume of manual adjustment work so time consuming that manually high resolution tracing clasts in the images in Illustrator using the most accurate fidelity selected for the pen, proved to be both faster and ensured no incorrect elements of the image were selected.*

- Lines 576-581: I do not fully follow the rationale here. You defined porosity based on seven assemblages (Figure 1), and you also have coral cover data as well as assemblage information. Why did you not use a weighted porosity approach, defined as:

Reef porosity(i) = $\sum(\text{cover}(i) \times \text{Porosity}(i))$; with the cover expressed from 0 to 1 instead of having percentage?

*This is an approach we had initially considered when seeking to better constrain assemblages and stacking porosity. Theoretically this would work well if one had completely monospecific fossil framework sequences to derive porosity values from e.g., only species x, or only species y present in a deposit. Very few fossil reef deposits are, however, entirely monospecific – there are a few exceptions of course – but across our dataset most deposits were not monospecific. Some *A. palmata* and *A. cervicornis* fossil sequences were (or came close) but most were mixed. Furthermore, because *A. palmata* and *A. cervicornis* are so rare or are now entirely absent from modern TWA reefs all contemporary sites in our assessment of RAPmax rates were characterised to varying degrees by mixed coral assemblages. These comprised combinations of massive/submassive/ encrusting or branched colonies. These mapped well to the fossil deposits we analysed and, for this reason, we used the most appropriate category for the mixed taxa present. We now make clear in the Fig 1 caption that these rates are associated with communities “dominated” by the specific species/or morphologies listed.*

Referee #2 (Remarks to the Author):

The manuscript entitled ‘Substantial water depth increases above coral reefs unavoidable beyond 2oC warming’ provides new and valuable insights into the potential for coral reefs in the Tropical Western Atlantic to keep up with future sea level rises. Overall, I thoroughly enjoyed reading the paper, it was well written with excellent figures and well referenced. I do have three areas where some more clarity or inclusion of discussion and/or detail might benefit the paper. These relate to: 1) reef framework porosity and sediment infilling, 2) future projections of carbonate budgets (and therefore related RAP calculations, and 3) accommodation space over reef flats (see below for further details). Regardless, the paper will be of broad interest and highlights the need for out of the box ideas around how we can upscale reef restoration efforts that increase a reefs carbonate budget and accretionary capacity. Hence, I recommend the paper for publication.

Comment 2.1. One of the key aspects that I applaud the authors for tackling is improvements to how reef accretionary potential (RAP) has been calculated in the past, which has involved a guesstimate of reef porosity. Here the authors, through the use of the fossil record, have devised a new and more quantifiable approach to assessing reef porosity – here termed stacking porosity. Stacking porosity varies between reefs due to variations in coral assemblages. My question here to the authors is that this technique would be somewhat limited to clearwater reefs given that we also know (albeit not very well quantified) of the importance of sediment infilling which can also ‘enhance’ reef accretionary rates by filling in the pores in the accreting reef structure. Hence there is potential here that these calculated porosity values which are from the fossil record – miss this important element that can facilitate reef accretion, and thereby underestimate accretionary potential.

Response: This is a very relevant point from the reviewer and one that would be especially relevant in many turbid reef type settings which we think underlies this question. The wider point here is that the values we present are most appropriate to the TWA and as pointed out, to clear water reefs in the region. In most such settings sediment accumulation occurs as a post-deposition infill – as old work by e.g., Scoffin & Tudhope showed. It does not seem to occur in the syndepositional way it has been shown to do in some e.g., GBR turbid-zone reefs, where accretion is driven by the combined effects of rapid coral growth and high-rate terrigenous sediment accumulation. Two points arise here. First, that it is probably reasonable to assume that any similar work (fossil framework analysis) being done that could select turbid-zone reef deposits would capture in those deposits the high proportion of sediment present. Presumably, this would therefore generate a very high stacking porosity

*value (unless there is a lot of post-depositional compaction). Second, we now make clear in the methods (where we have some word space) that these values are primarily TWA specific. These values are probably broadly Indo-Pacific compatible but there would be merit in similar analysis being done on Indo-Pacific fossil sequences (see new **Line 588 on**).*

Comment 2.2. My second question relates to future predictions of net carbonate production, which the authors have calculated using future projections of temperature and pH models. I understand why the authors have used this approach but looking over table S4 – declines in coral cover, calcification rates etc are seemingly generic (i.e. the same over time), and if there is one thing I've learnt from working on coral reefs is of the highly heterogeneous nature within let alone between reefs. So to suggest that all reefs will respond to the same degree I think is a huge simplification of how reefs will respond to these future stressors. I also understand that getting these types of site specific data is hard – but I would like to see some acknowledgement that not all reefs are the same for both internal and external reasons.

Response: We suspect that there may have been a slight misreading of Table S4 – but apologies if we have misread this comment. In fact, the rates of coral cover decline, impacts on calcification, bioerosion etc we use in our projections all change from present to 2100 and by increasing amounts under increasingly extreme SSP scenarios. Each location in Table S4 reads from left to right by time period and scenario, and the values increase/change accordingly. Geographically, the projections were calculated from grid sizes between 25km and 100km nominal resolution (depending on the specific climate output model) and were then applied to our data at the site level. This is currently the highest resolution projection data we can realistically generate. In SI table 4 projections are reported as the means for each sector within a region - there were sometimes some slight differences between sites in each sector and they were applied in the actual projections as appropriate. Speaking to the review point this is, as the reviewer notes, critical because reefs across regions will divergently experience effects from e.g., SST and OA changes. We have thus applied relevant change rates to each site from the gridded models although the effects of these between different sub-areas of our study locations are often subtle (but where they do differ these were used). The central point here is that we have not assumed all reefs will be exposed in the same way across the TWA.

Comment 2.3. Finally, I always wonder – are there reefs (which may not include these reefs here as I am less familiar with Caribbean reef systems) – where an increase in sea level be a good thing for carbonate production? That is, for those reef flats (which can be vast areas in the 10 km²), which are essentially at a hiatus and corals living in these environments are relatively hard core i.e. they are dealing with extreme conditions and yet surviving. Could an increase in SL allow these corals to grow more, there by increasing carbonate production and therefore reef accretionary potential. Food for thought. Perhaps not easily woven in here, or perhaps something to add in as a discussion point???

*Response: We agree and we refer back here to our response to comment 1.1 – but this comment relates to the very different reef geomorphic states defining the TWA compared to the Indo-Pacific, and especially the Pacific. The former have very narrow, near SL crests which are at or about 0.5 m below water at most. In contrast, many Pacific reefs have wide expansive reef flats that are semi or completely emergent at low tide. As we outline in our response to reviewer 1 the potential geomorphic “capacity” and lateral habitat space for renewed reef colonisation under SLR is thus very different between these geographies. As above we have added a brief comment on this differential re-colonisation potential in the Pacific. **New Lines 249 on, and 264 on.***

Referee #3 (Remarks to the Author):

The manuscript by Perry and collaborators presents a valuable assessment of the impacts of sea-level rise (SLR) on Caribbean coral reef environments and highlights the importance of proactive measures to mitigate some of the negative effects. Notably, the modelling approach includes optimistic scenarios in which strong and aggressive restoration efforts are implemented. The study offers a novel perspective by integrating a reassessment of geological data with contemporary coral growth and cover information, along with climate scenario modelling. Overall, the manuscript is well written, the topic is timely and relevant, and the analysis is robust. I therefore support its publication, pending consideration of a few minor comments.

Comment 3.1. One of the central findings appears to be the projected near-total collapse of reef accretion by 2040 across all scenarios. To what extent has the potential for coral acclimatization, e.g. genetic selection, or modulation by climatic modes of variability been considered in these projections? Line 339 mentions that acclimatization/adaptation could occur — has this been incorporated into the model, or is it acknowledged as an uncertainty?

*Response: The short answer here is that we have not integrated or factored here for coral acclimatization. As the reviewer notes this may be one pathway that might enable higher rates of coral survival (either of naturally recovery populations or of out-planted corals). There is probably not yet enough clear evidence of this occurring at scale. It would theoretically be possible to impose e.g., a set temperature tolerance increase as an additional run in our projections as previous work has done (e.g., Cornwall et al. 2023), but this would at best remain a projection with very high levels of uncertainty given we have poor constraints on the pace of adaptation (if it occurs) and how this would vary with species. We now make clear in the text/methods that this is not factored for in our projections and that there is uncertainty around the speed of any benefits – **New Line 291 and 629 on.***

Comment 3.2. Additionally, has the changes in physical oceanography associated to increased water depth been considered in the models? For instance, might it enhance water exchange with cooler oceanic waters or facilitate the removal of nutrients and pollutants, thereby mitigating some stressors? Conversely, could it worsen conditions by increasing water column stratification and reducing vertical mixing?

*Response: This is a very interesting comment. Again, short answer is no – not included here and sits somewhat outside the scope of what we can realistically consider here empirically without high resolution bathymetry and physical oceanography data. We are not aware of work showing that 0.5-1.5 m water depth increases might substantially increase deeper and cooler water exchanges across the tops of reefs. It is realistic, however, to image that across reef water exchange may well increase (as a couple of modelling studies we are aware of have shown). As the reviewer notes this could help nutrient flushing – an issue in many TWA reefs, although of course conversely changing across reef wave energy regimes could also enhance sediment influx from shoreline erosion. It is probably also reasonable to assume that a further implication will be impacts on nearshore/lagoon benthic habitats. It is not clear to us how we could realistically include these factors but as per the first comment here a brief comment in the text about possible changes in water exchange and ecologies as water depths increase is definitely warranted. See addition to abstract (**new line 50**) and in text (**new line 240 onwards**).*

Line 74 – This statement is somewhat unclear. Since SLR is itself a consequence of climate change, it would be helpful to clarify here that you are referring to the direct (non-SLR) impacts of climate change.

*Response: SLR per se is not a major threat to reefs – or at least not at the magnitudes projected over the next century – in the sense that they will not drown out reefs, but we fully understand and agree with the point made here by the reviewer and have modified the text accordingly. **Line 79.***

Figures 1 and 2 – RAP is defined as reef accretion rates rather than reef accretion potential. Please clarify the terminology to ensure consistency and prevent confusion.

Response: A valid spot. Corrected. Thank you.